# C3R: Channel Conditioned Cell Representations for unified evaluation in microscopy imaging

## Abstract

Immunohistochemical (IHC) images reveal detailed information about structures and functions at the subcellular level. However, unlike RGB images, IHC datasets pose challenges for deep learning models due to their inconsistencies in channel count and configuration, stemming from varying staining protocols across laboratories and studies. Although existing approaches build channel-adaptive models, they do not perform zero-shot evaluation across IHC datasets with unseen channel configurations. To address this, we first introduce a structured view of cellular image channels by grouping them into either context or concept, where we treat the context channels as a reference to the concept channels in the image. We leverage this view to propose Channel Conditioned Cell Representations (C3R), a framework that learns representations that transfers well to both in-distribution (ID) and out-of-distribution (OOD) datasets which contain same and different channel configurations, respectively. C3R is a two-fold framework comprising a channel-adaptive encoder architecture and a masked knowledge distillation training strategy, both built around the context-concept principle. We find that C3R outperforms existing benchmarks on both ID and OOD tasks, while yielding state-of-the-art results on CHAMMI-ZS; a zero-shot-style adaptation of the CHAMMI benchmark. Our method opens a new pathway for cross-dataset generalization between IHC datasets, with no need for retraining on unseen channel configurations.

## 1 Introduction

Immunohistochemical (IHC) images reveal subcellular patterns that provide rich information about cell structure and function, enabling machine learning methods perform tasks such as disease prediction, subcellular localization, and prediction of drug perturbation outcomes. Recently, self-supervised learning (SSL) on IHC datasets followed by evaluation on in-distribution (ID) and out-of-distribution (OOD) tasks within or across IHC datasets has become the standard paradigm for assessing vision foundation models in this domain (Bray et al., 2016; Caicedo et al., 2022; Doron et al., 2023; Gupta et al., 2024; Kim et al., 2025).

However, learning strong image representations via SSL that are transferable across IHC datasets remains challenging. Different staining protocols produce different channel configurations across datasets, while typical image encoders assume a fixed number of channels. This makes training and evaluating models across multiple channel configurations non-trivial. Prior studies address multi-channel training across datasets (Chen et al., 2023; Bao et al., 2024; Pham and Plummer, 2024), but provide no mechanism for OOD evaluation on unseen channel configurations, limiting their evaluations to configurations observed during training.

To address this, we identify an intrinsic separation of channels in IHC datasets that reflects laboratory-specific experimental design choices (Thul et al., 2017; Chandrasekaran et al., 2024; Viana et al., 2023). We find that certain 'context' channels, such as the Nucleus or ER, provide a structural reference of the cell, and are visually consistent across images in a dataset. In contrast, 'concept' channels, such as Protein, capture experiment-specific semantic information, and are meant to be interpreted in relation to the context channels (Fig. 1).

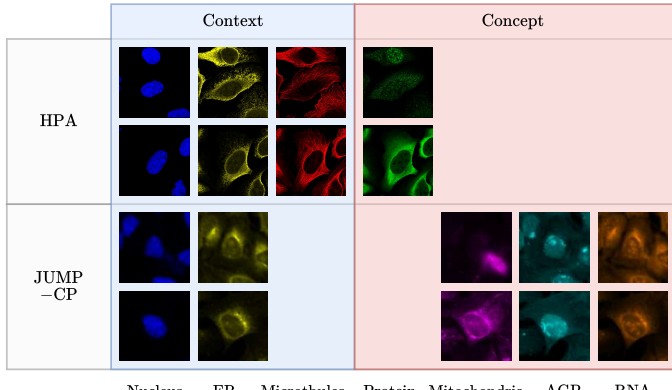

Figure 1: The intrinsic separation of channels in IHC datasets. Context channels (left columns) serve as structural references and tend to exhibit high visual consistency across cells. Concept channels (right columns) capture more variable, experiment-specific features, and exhibit greater semantic diversity across instances.

An analogy to context and concept can be drawn from natural images, where the context corresponds to the environment and concept to the object of interest. Unlike natural images, IHC datasets explicitly separate this information into distinct channels, each carrying either contextual or conceptual signals. We refer to this as the 'context–concept principle' in IHC channels, and take a step towards leveraging it to learn high-quality, transferable representations across IHC datasets. To this end, we propose **Channel-Conditioned-Cell Representations (C3R)**, a two-fold framework that combines (i) an architectural design with strong representational capacity and support for OOD evaluation, and (ii) a pre-training method that further strengthens IHC image representations.

We begin from the observation that context and concept channel groups carry semantically distinct information. Based on this, we propose the Context–Concept Encoder (CCE), a network architecture that acts as an information handler for distinct context and concept groups. The CCE model learns separate group wise intermediate features to capture within-group structure, and then integrates them to learn a unified final representation. During inference, unobserved channels are manually assigned to either group and processed without additional training. We find that the group-wise information handling in CCE yields strong representations over baseline methods, while the group assignment using CCE makes systematic OOD evaluation possible.

Next, we further leverage the structural coherence of context channels and the dependence of concept channels on context. We propose Masked Context Distillation (MCD), a momentum-based pre-training strategy that enhances representations for IHC images. MCD introduces a distillation loss that regulates how much context the concept channels access when forming the overall representation. This encourages concept channels to contribute effectively to the overall representation even with limited context, yielding stronger, concept-driven IHC image representations.

Together, CCE and MCD form the C3R framework, enabling both systematic OOD evaluation and improved representation learning across diverse IHC datasets. We evaluate C3R on downstream tasks on HPA (ID) (Thul et al., 2017) and JUMP-CP (OOD) (Chandrasekaran et al., 2023) datasets, where C3R outperforms existing ID and OOD baselines. We also evaluate C3R on CHAMMI-ZS, a zero-shot style adaptation of the CHAMMI benchmark (Chen et al., 2023). In addition to the original CHAMMI benchmark that evaluates generalization to novel semantic differences within IHC datasets, CHAMMI-ZS evaluates generalization to unobserved channels across IHC datasets. C3R achieves state-of-the-art performance on CHAMMI-ZS, surpassing existing channel-adaptive and channel-agnostic methods.

## 2 RELATED WORK

**Representation learning for IHC images.** Representation learning using SSL for cell images ranges from fully unsupervised to weakly supervised approaches. While some focus on scaling

(Kenyon-Dean et al., 2024; Lorenci et al., 2025; Kraus et al., 2024), we focus on architectural and SSL design choices. In fully unsupervised methods, discriminative approaches such as SimCLR, DINO, and iBOT (Chen et al., 2020; Caron et al., 2021; Zhou et al., 2021) have been shown to outperform reconstruction-based methods (Xun et al., 2024; Murthy et al., 2024) in IHC datasets. As such, DINO4Cells (Doron et al., 2023) shows that DINO learns unbiased morphology features, learning useful cell structure without labels. In addition to fully unsupervised SSL, weak supervision introduces biological signals to further improve representations. Examples of these signals include protein identity, screen-derived labels, or RNA-seq data (Kobayashi et al., 2022; Yao et al., 2024; Fu et al., 2024). SubCell (Gupta et al., 2024) uses a supervised contrastive loss with antibody-stained cells, achieving strong results on ID and OOD datasets. However, they re-train models to match OOD datasets, which limits direct transfer to unobserved channels. We use DINO4Cells (Doron et al., 2023) and SubCell (Gupta et al., 2024) as benchmarks, since they are trained and tested on HPA (Thul et al., 2017) and JUMP-CP (Chandrasekaran et al., 2023), but require retraining to handle OOD channels. In contrast, our models are evaluated without retraining.

**The context-concept principle in IHC images.** IHC datasets consistently separate channels into those that provide structural context and those that capture task-specific concepts. In the HPA dataset, each sample includes the protein of interest alongside three reference markers (Nucleus, Microtubules, and Endoplasmic Reticulum (ER)) which serve as structural landmarks (Thul et al., 2017). Similarly, in JUMP-CP, the DNA (Nucleus) and ER channels act as positional references while the remaining channels (RNA, AGP, Mitochondria) capture perturbation-induced morphological phenotypes (Chandrasekaran et al., 2023). In WTC-11, the Nucleus provides the primary reference signal, while and Membrane and Protein channels contribute auxiliary information (Viana et al., 2023). These examples show that certain channels act as stable context for cellular localization, while others provide variable, concept-driven information relevant to the biological task.

**Vision transformers for multi-channel imaging.** Handling multi-channel data has previously been studied outside biology, for example in climate modelling (Herruzo et al., 2021; Nguyen et al., 2023). For IHC datasets, the CHAMMI benchmark (Chen et al., 2023) evaluates cross-dataset multi-channel training with over 220,000 single-cell images from three sources. However, the CHAMMI benchmark does not evaluate generalization to unobserved channels. Architectures such as ChAdaViT (Bourriez et al., 2024), ChannelViT (Bao et al., 2024), and DiChaViT (Pham and Plummer, 2024) tokenize channels separately, assign channel-specific parameters and use sampling or regularization to promote channel and token diversity. However, these methods remain tied to channels seen during training, and they scale poorly with the number of channels due to the large number of tokens. As an alternative, single-channel approaches (Lorenci et al., 2025; Xun et al., 2024; Murthy et al., 2024; Lian et al., 2025) process each channel independently. Although these models can be directly evaluated on unseen datasets without further training, they fail to capture inter-channel dependencies. Nonetheless, we include a single-channel approach in our evaluation for completeness.

## 3    METHOD

We first revisit the Context-Concept principle in the setting of IHC images, providing background and motivation for C3R. Then, we introduce C3R, our proposed method for learning cell-level features conditioned on this principle. The C3R pre-training code is available at `https://anonymous.4open.science/r/C3R-5015`.

### 3.1    INVESTIGATING THE CONTEXT-CONCEPT PRINCIPLE IN IHC DATASETS

The core idea of the context-concept principle between IHC channels exists in literature (Thul et al., 2017; Viana et al., 2023; Chandrasekaran et al., 2023). Our motivation for C3R is based on this principle, and through this, the hypothesis that channels within these groups encode distinct types of information. As shown in Fig. 1, context channels already appear visually consistent across samples. Given this, we quantitatively verify two aspects that support the hypothesis of group distinction; first, that images of a single channel are generally similar across instances in the dataset, and second, that channels that belong to the concept group exhibit higher variability than context channels, as they are the primary focus for downstream tasks.

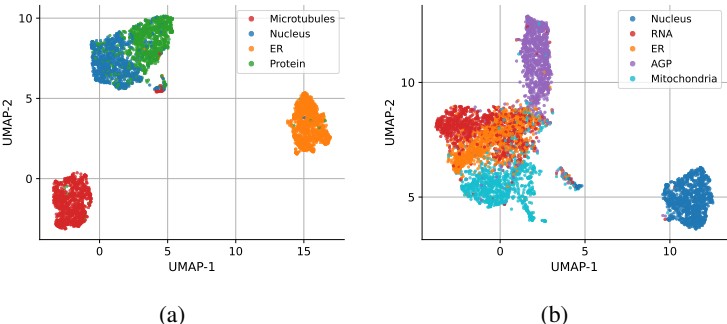

Table 1: Average per-group Parity $P_c$ and entropy $H_c$. For individual channels, see Appendix B.1.

|  | $P_c$ | $H_c$ |
|---|---|---|
| **HPA** | | |
| Context | **0.935** | 0.390 |
| Concept | 0.759 | **1.025** |
| **JUMP-CP** | | |
| Context | **0.716** | 0.934 |
| Concept | 0.362 | **1.792** |

Figure 2: Visualization of channel-wise features in 2D space using UMAP on (a) HPA and (b) JUMP-CP. Most channels show clear inter-channel separation and intra-channel similarity.

**IHC channels vary more across channels than instances.** To first validate that IHC channels capture visually similar patterns across instances, we compare channel-wise features across datasets. To do this, we generate global feature representations of individual channels from $N = 1000$ random instances in the HPA and JUMP-CP datasets, which contain 4 and 5 channels, respectively. We use a natural image checkpoint (Oquab et al., 2023) to prevent inducing a domain-specific bias when computing the representations, as we focus on evaluating the visual patterns of each channel. To match the model input, we repeat the single-channel image thrice along the channel dimension to emulate an RGB image. We first visualize the computed features in 2D using UMAP (McInnes et al., 2018), and as shown in Fig. 2, individual channel representations are distinct in the feature space and show minimal overlap. However, Fig. 2 does not provide a clear visual indication that concept channels show higher variability than context channels. Therefore, to investigate the second point, we quantitatively evaluate the group distinction and find that the context channels are visually similar across instances as opposed to concept channels.

**Context channels are more similar across instances than concept channels.** To quantify the similarity of instances within groups, we first cluster all features of all channels in a given dataset using K-Means (Lloyd, 1982). We use $K = 4$ clusters on HPA, and $K = 5$ clusters on JUMP-CP as they have 4 and 5 channels respectively. Perfect clustering of channel features into $K$ separate clusters would imply that channels carry independent information that is constant across instances. In contrast, a channel falling into more than one cluster would imply that the channel shows some variability across instances. We quantify this as follows.

In a given dataset, for a given channel $c$, we compute the relative occurrence of that channel's features across the clusters as,

$$\mathbf{p}_c = \{p_c^k \mid k \in \{0, \ldots, K-1\}\}, \quad \text{with} \quad p_c^k = \frac{1}{|\mathcal{I}_c|} \sum_{n \in \mathcal{I}_c} \mathbb{1}(c_n = k) \tag{1}$$

where $\mathcal{I}_c = \{n \mid c_n = c\}$ is the set indices of feature samples that belong to channel $c$. Hence, $p_k^c$ returns the fraction of samples of channel $c$ that fall into cluster $k$. Each channel's 'assigned' cluster becomes the cluster in which its features appear most frequently.

We then compute the parity $P_c$ and entropy $H_c$ of each channel $c$. The parity $P_c$ measures the concentration of a single channel $c$ in its assigned cluster, whereas the entropy $H_c$ measures the spread of channel $c$ across all clusters (Shannon, 1948). High parity and low entropy for a given channel indicates a tight spread of the channel features in the feature space, and vise versa for low parity and high entropy. We compute $P_c$ and $H_c$ as,

$$P_c = \max(\mathbf{p}_c) \quad \text{and} \quad H_c = - \sum_{k=0}^{K-1} (p_c^k) \cdot \log_2(p_c^k) \quad . \tag{2}$$

Table 1 shows the computed parity $P_c$ and entropy $H_c$ for each group (context or concept) averaged over the channels in that group for each dataset. For each dataset, we observe a high parity for

the context channels vs. the concept channels, and high entropy for the concept channels vs. the context channels. This validates the second aspect of our hypothesis: context channels are generally more consistent across instances, whereas concept channels exhibit high variability. This suggests that the two groups convey distinct types of information, with context channels capturing structural information and concept channels capturing semantic, task-relevant information.

## 3.2 C3R: CHANNEL CONDITIONED CELL REPRESENTATIONS

Based on the context-concept principle validated in Section 3.1, we propose Channel Conditioned Cell Representations (C3R). C3R includes two main components: the Context-Concept Encoder (CCE) and the Masked Context Distillation (MCD) training strategy. The two components when combined, encourages distinct feature learning for each context and concept group, while encouraging the context channels to act as a reference for the concept. We implement C3R using a Vision Transformer (ViT) backbone and use iBOT as our base pre-training pipeline (Zhou et al., 2021; Dosovitskiy et al., 2021).

To describe CCE and MCD, we first define the input $x$ as a multi-channel IHC image, viewed as two groups of images, where $C_1, C_2$ are number of channels in the context and concept groups, and $(h, w)$ are the spatial dimensions. The group assignments for each dataset can be found in Table 7 in the supplementary material.

$$x = [x_{c1}, x_{c2}], \quad x_{c1} \in \mathbb{R}^{C_1 \times h \times w}, \quad x_{c2} \in \mathbb{R}^{C_2 \times h \times w}. \tag{3}$$

**Context-Concept Encoder (CCE).** We design CCE to first construct separate intermediate representations for context and concept channel groups, then combine them to model inter-group dependencies and form a global representation. Leveraging the inherent distinction between these groups outlined in Section 3.1, CCE encodes the context and concept groups independently up to a certain depth. Since the context-concept distinction is consistent across datasets, the model serves as an information handler of these groups in an OOD setting with novel channels.

The CCE architecture comprises group-wise convolutional stems and lightweight encoder layers for processing context and concept-specific inputs ($h_{c1}, h_{c2}$ for stems; $f_{c1}, f_{c2}$ for encoders), a combiner function $g$ for merging the group-wise representations, and joint encoder layers $f_s$ for further joint processing, as illustrated in Fig. 3a. A high-level overview is provided here, while full architectural specifications of all components ($h_c, f_c, g, f_s$) are detailed in Appendix A.1.

Given the input $x = [x_{c1}, x_{c2}]$, channels in $x_{c1}$ and $x_{c2}$ are tokenized by group-specific stems,

$$\tilde{x}_{c1}^i = h_{c1}(x_{c1}^i) \in \mathbb{R}^{N \times d}, \quad i = 1, \dots, C_1, \tag{4}$$

$$\tilde{x}_{c2}^j = h_{c2}(x_{c2}^j) \in \mathbb{R}^{N \times d}, \quad j = 1, \dots, C_2, \tag{5}$$

where $d$ is the output dimensionality of $h_{c1}$ and $h_{c2}$, and $N$ is the number of tokens. Each tokenized channel is then encoded independently via group-specific branched lightweight encoders,

$$\hat{x}_{c1}^i = f_{c1}(\tilde{x}_{c1}^i) \in \mathbb{R}^{N \times d}, \quad i = 1, \dots, C_1, \tag{6}$$

$$\hat{x}_{c2}^j = f_{c2}(\tilde{x}_{c2}^j) \in \mathbb{R}^{N \times d}, \quad j = 1, \dots, C_2, \tag{7}$$

and then combined by a function $g$ (see Appendix A.1 for implementation details), before passing through joint encoder layers to obtain the final output $y$,

$$\hat{x} = g\left(\{\hat{x}_{c1}\}_{i=0}^{C_1}, \{\hat{x}_{c2}\}_{j=0}^{C_2}\right) \in \mathbb{R}^{N \times D}, \tag{8}$$

$$y = f_s(\hat{x}) \in \mathbb{R}^D. \tag{9}$$

Typically, $D = 2d$, where $D$ is the embedding dimension of the baseline encoder, and the layer depths of $f_c$ and $f_s$ are set to match the overall parameter count of baselines. This ensures fair comparison by parameter and FLOP count versus comparative methods. However, in contrast to the comparative methods, the distinction of groups learned by CCE to yield the final representation $y$ is directly transferable to other datasets with unobserved channels.

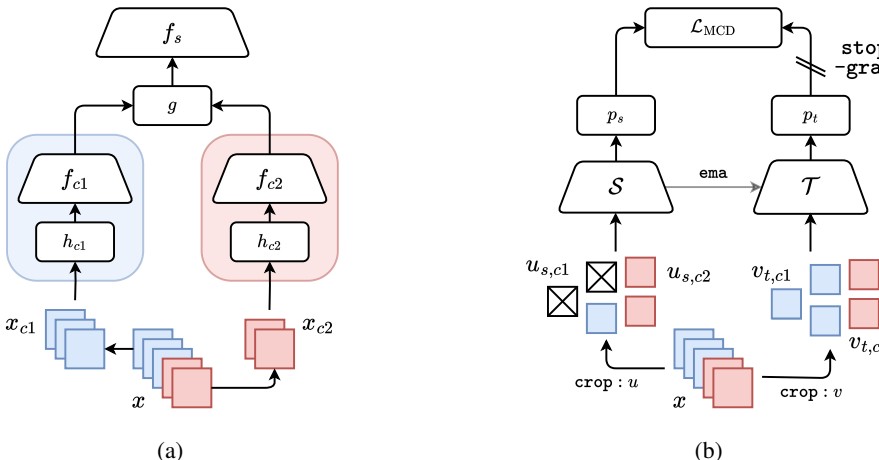

(a)            (b)

Figure 3: (a) Context-Concept Encoder: The input channels are separated into context and concept, where each group is processed independently through their respective $h_c$ and $f_c$ layers. The two group-wise representations are then combined and passed through joint encoder layers $f_s$. (b) Masked Context Distillation: During training, the student encoder $\mathcal{S}$ randomly samples a subset of context channels prior to the forward pass, while the teacher encoder $\mathcal{T}$ passes the full set of context channels. The loss is computed between the context-masked student representation and the dense teacher representation.

**Masked context distillation (MCD).** MCD is a pre-training strategy that governs the interaction between context and concept channels during training. Based on the structural coherence of context channels across samples as observed in Section 3.1, using MCD, we encourage the concept channels of the model to infer from limited context, building robust global representations.

The core pretraining pipeline of our work is based on iBOT (Zhou et al., 2021), which contains masked patch prediction and student-teacher distillation. However, unlike existing distillation methods in DINO and iBOT, the student signal in MCD differs from the teacher signal not only by cropping and augmentation, but also through a novel context channel sampling strategy that allows for conditioning of the concept channel through limited context. Fig. 3b shows the schematic of the distillation strategy in MCD.

We define the student and teacher networks as $\mathcal{S}$ and $\mathcal{T}$, and each network contains a projection head defined as $p_s$ and $p_t$, respectively. Given the input $x = [x_{c1}, x_{c2}]$, we draw two augmented views $u = [u_{c1}, u_{c2}]$ and $v = [v_{c1}, v_{c2}]$.

Assuming crop $u$ is passed to the student network and $v$ to the teacher, we randomly sample without replacement $c$ channels from the context group $u_{c1}$, where $1 \leq c \leq C_1$. Conversely, when $u$ is passed to the teacher and $v$ to the student, sampling is instead applied to the context group $v_{c1}$. Given student crop $u$ and teacher crop $v$, the student and teacher inputs are then defined as,

$$u_{s,c1-\texttt{smp}} = \texttt{sample}(u_{c1}, c) \in \mathbb{R}^{c \times h \times w}, \quad \text{where } 1 \leq c \leq C_1, \tag{10}$$

$$u_s = [u_{c1-\texttt{smp}}, u_{c2}], \quad v_t = [v_{c1}, v_{c2}]. \tag{11}$$

$u_s$ and $v_t$ are then passed through student and teacher encoders $\mathcal{S}$ and $\mathcal{T}$ to obtain feature representations, and then passed through respective projection heads $p_s$ and $p_t$ to obtain $z_s$ and $z_t$. The Masked Context Distillation loss then is computed as the KL-divergence between them.

$$y_s = \mathcal{S}(u_s), \quad y_t = \mathcal{T}(v_t), \tag{12}$$

$$z_s = p_s(y_s), \quad z_t = p_t(y_t) \tag{13}$$

$$\mathcal{L}_{\text{MCD}} = \mathcal{L}_{\text{KL}}(z_s, z_t). \tag{14}$$

The pseudo-code for MCD and the effects of channel sampling rates can be found in Appendix A.2 and Appendix B.3 respectively. By the separate-and-joint processing of context and concept channels through CCE, and the conditioned learning of the context-concept relationship through MCD, our method yields strong image representations and generalize well to unseen IHC datasets.

## 4 RESULTS

### 4.1 EXPERIMENTS

**Pre-training and evaluation** All models (except SubCell and DINO4Cells for which we use their pre-trained checkpoints) are pre-trained on the HPA training set ($\approx 800k$ images) using iBOT (Zhou et al., 2021) with the SubCell antibody loss (Gupta et al., 2024). Adding to comparative methods, we train three baselines: **Base-HPA** (4-channel HPA), **Base-CP** (3-channel HPA, excluding Microtubules), and **Base-SC** (single-channel HPA, one channel per forward pass). ID evaluation (**HPA-Loc**) is performed on the HPA test set for 19,31-class protein localization, reported in mAP.

OOD evaluation (cross-dataset evaluation with frozen features) is conducted on JUMP-CP and the CHAMMI benchmark dataset (containing images from WTC-11, HPA, and JUMP). On JUMP-CP, we test two settings with frozen features: (i) **JUMP-Ret**: zero-shot compound replicate retrieval (matching drug perturbations between cells); and (ii) **JUMP-TRex**: task-specific feature adaptation, following TRex (Husain et al., 2025), where linear classifiers are trained with task-specific losses on frozen IHC features for compound replicate matching and mechanism-of-action (MoA) prediction (a task that yields random performance in zero-shot).

On the CHAMMI benchmark dataset, we introduce **CHAMMI-ZS**, a zero-shot variant of the original benchmark designed to test generalization to unseen channels. As in the original benchmark, joint training is performed on three IHC datasets and evaluated across nine tasks. However, CHAMMI-ZS trains a 2-layer MLP on frozen image representations. This prevents encoder or stem network fine-tuning like in CHAMMI, thereby preventing the HPA-pretrained encoder from adapting weights to new channel configurations, and as a result emulating a zero-shot setting. A comparison between downstream evaluation tasks can be found in Table 13 in the supplementary material.

**Comparisons** We pre-train C3R with ViT-S and ViT-B (with adjusted layer depths) and compare against Base-HPA, Base-CP, Base-SC, DINO4Cells (Doron et al., 2023), SubCell (Gupta et al., 2024), ChannelViT (Bao et al., 2024), and DiChaViT (Pham and Plummer, 2024). We use official repositories and model checkpoints when available and otherwise re-implement methods under our setting. We set the depths of $f_c$ and $f_s$ to be 2 and 11 for C3R, and the choice of depths is investigated in Appendix B.2. Further implementation details of all methods can be found in Appendix C.2 in the supplementary material.

### 4.2 BENCHMARK RESULTS

**HPA-Loc, JUMP-Ret and CHAMMI-ZS** Table 2 summarizes the results on the HPA protein localization, JUMP-CP zero-shot compound retrieval, and CHAMMI-ZS benchmarks. The results for CHAMMI-ZS containing all tasks can be found in Appendix B.4. From Table 2, we find that C3R outperforms all methods except Base-CP with ViT-S, which is pre-trained on HPA with a channel configuration matched to JUMP-CP. Without any target-specific pre-training, C3R achieves nearly the same performance as Base-CP in this setting, and surpasses all other methods across both ViT-S and ViT-B. We observe significant improvements on CHAMMI-ZS and HPA-Loc, while on JUMP-Ret, C3R matches or exceeds Base-CP and substantially outperforms the remaining methods.

**JUMP-TRex** Table 3 presents results on JUMP-CP using TRex with ViT-B trained on A549 and U2OS cell lines separately, and then on both cell lines jointly. C3R outperforms all methods on all experiments on compound matching, while it outperforms all methods on 2/3 experiments on MoA identification. It is worth noting that Base-CP, Dino4Cells and SubCell are pre-trained to match the target channel configuration, while C3R is not. While zero-shot retrieval capabilities are valuable for broad applicability without retraining, C3R's advantages also extend to scenarios where features are explicitly adapted to subsets of the target dataset via linear probing.

### 4.3 ANALYSIS

**The contribution of each component of C3R.** Table 4 shows the incremental contribution of each component in C3R over the baseline ViT. We first replace the standard convolutional stem of the ViT with per-group stems $h_{c1}, h_{c2}$. Here, the combiner $g$ from Section 3.2 is applied directly

Table 2: Results on HPA-Loc, JUMP-Ret (OOD), and CHAMMI-ZS benchmarks. For CHAMMI-ZS, we report the ID and OOD average scores, and overall CHAMMI Performance Score (CPS). *: re-trained from scratch to match the JUMP-CP channel configuration. //: unsuitable for evaluation due to channel mismatch.

| Encoder | Method | HPA-Loc | | JUMP-Ret | | CHAMMI-ZS (F1) | | |
|---|---|---|---|---|---|---|---|---|
| | | mAP-31 | mAP-19 | mAP | kNN | ID | OOD | CPS |
| ViT-S | Base-HPA | 0.505 | 0.686 | // | // | // | // | // |
| | Base-CP* | // | // | **0.355** | 0.507 | // | // | // |
| | Base-SC | 0.380 | 0.528 | 0.327 | 0.457 | 0.812 | 0.427 | 0.474 |
| | ChannelViT | 0.438 | 0.602 | 0.345 | 0.503 | 0.852 | 0.423 | 0.472 |
| | DiChaViT | 0.429 | 0.590 | 0.343 | 0.494 | 0.812 | 0.410 | 0.459 |
| | C3R | **0.536** | **0.722** | 0.354 | **0.518** | **0.861** | **0.485** | **0.543** |
| ViT-B | Base-HPA | 0.515 | 0.698 | // | // | // | // | // |
| | Base-CP* | // | // | 0.355 | 0.513 | // | // | // |
| | Base-SC | 0.385 | 0.528 | 0.339 | 0.473 | 0.797 | 0.423 | 0.459 |
| | DINO4Cells* | 0.508 | 0.683 | 0.339 | 0.509 | // | // | // |
| | SubCell* | 0.519 | 0.695 | 0.350 | 0.514 | // | // | // |
| | C3R | **0.548** | **0.737** | **0.363** | **0.530** | **0.873** | **0.468** | **0.522** |

Table 3: Results on JUMP-TRex, trained on cell lines A549 and U2OS on ViT-B. *: re-trained to match target channel configuration.

| Method | A549 | | U2OS | | A549+U2OS | |
|---|---|---|---|---|---|---|
| | Cmpd | MoA | Cmpd | MoA | Cmpd | MoA |
| Base-CP* | 0.427 | 0.306 | 0.335 | 0.276 | 0.321 | 0.281 |
| Base-SC | 0.381 | 0.234 | 0.276 | 0.211 | 0.268 | 0.231 |
| DINO4Cells* | 0.439 | **0.348** | 0.314 | 0.287 | 0.310 | 0.329 |
| SubCell* | 0.424 | 0.291 | 0.334 | 0.260 | 0.323 | 0.267 |
| C3R | **0.444** | 0.346 | **0.345** | **0.298** | **0.325** | **0.341** |

on the outputs of $h_c$. We observe that $h_{c1}, h_{c2}$ alone improves ID performance on HPA. However, when applied to JUMP-CP, it outperforms Base-SC, ChannelViT, and DiChaViT (methods with workarounds for unseen channels, results shown in Table 2), but falls short of our OOD baseline set by re-training an encoder to match JUMP-CP's channel configuration (Base-CP). We suspect that while per-group convolutional stems aid low-level feature extraction, they fail to learn a context-concept distinction, limiting transfer to the OOD dataset. Introducing $f_{c1}, f_{c2}$ before $g$, thereby processing input groups independently for several layers, significantly improves OOD performance, matching or surpassing Base-CP, suggesting this distinction is learned in these branched encoder layers. We experimentally validate this later in this section.

Applying masked context distillation (MCD), where some context channels are dropped during training and the student network benefits from the teacher's full context channel representation, yields significant improvements in ID performance. However, we find that the ID performance boost obtained by MCD does not translate to the JUMP-Ret task. Nevertheless, we find improvements with MCD in both CHAMMI-ZS and JUMP-TRex (see Appendix B.3). We suspect that some learnable adaptation is necessary to extract strong representations from these frozen features.

**Experimental validation of group-specific learning.** To further validate the hypothesis that the branched encoders $fc1$ and $fc2$ encode distinct context-concept information, we deliberately flip group assignments during OOD evaluation. A significant drop in performance would suggest that $f_{c1}$, $f_{c2}$ learn group-specific distinct information which is passed on to the OOD dataset. Conversely, if the performance remains similar, it would indicate that $f_{c1}$, $f_{c2}$ are group-agnostic, implying that the performance increase is simply a result of channel separation during training. The results in Fig. 4 support the hypothesis, as flipping consistently reduces performance. Interestingly, the performance

Table 4: The effect of each component in C3R. *: pre-trained to match the JUMP-CP channel configuration. $h_c$: grouped stems, $f_c$: branched encoders, 'MCD': context distillation. //: unsuitable for evaluation due to channel mismatch.

| Method | ViT-S | | | | ViT-B | | | |
| --- | --- | --- | --- | --- | --- | --- | --- | --- |
| | HPA-Loc | | JUMP-Ret | | HPA-Loc | | JUMP-Ret | |
| | mAP-31 | mAP-19 | mAP | kNN | mAP-31 | mAP-19 | mAP | kNN |
| Base-HPA | 0.505 | 0.686 | // | // | 0.515 | 0.698 | // | // |
| Base-CP* | // | // | 0.355 | 0.507 | // | // | 0.355 | 0.513 |
| Base-SC | 0.380 | 0.528 | 0.327 | 0.457 | 0.385 | 0.528 | 0.339 | 0.473 |
| $h_c$ | 0.523 | 0.700 | 0.347 | 0.510 | 0.529 | 0.710 | 0.344 | 0.508 |
| $h_c + f_c$ | 0.520 | 0.705 | 0.351 | **0.529** | 0.531 | 0.716 | 0.358 | **0.532** |
| $h_c + f_c + \mathrm{MCD}$ | **0.535** | **0.725** | **0.354** | 0.518 | **0.548** | **0.737** | **0.363** | 0.530 |

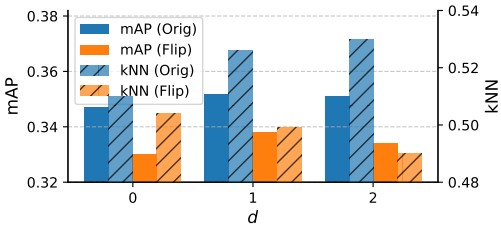

Figure 4: Effects of group switching assignments. $d$: layers per each $f_c$. When $d = 0$, groups are combined directly after $h_c$ stems.

Table 5: Choice of channel dropping for MCD. Using ViT-S. $\mathcal{S}, \mathcal{T}$: student, teacher.

| Drop | HPA-Loc | | JUMP-Ret | |
| --- | --- | --- | --- | --- |
| | AP-31 | AP-19 | mAP | kNN |
| ChannelViT | 0.438 | 0.602 | 0.345 | 0.503 |
| DiChaViT | 0.429 | 0.590 | 0.343 | 0.494 |
| None | 0.519 | 0.702 | 0.351 | **0.530** |
| $\mathcal{S}$ only | **0.536** | 0.722 | **0.354** | 0.518 |
| $\mathcal{S} + \mathcal{T}$ | 0.533 | **0.724** | 0.347 | 0.504 |

drop grows with deeper $f_{c1}$ and $f_{c2}$ layers, indicating that separating the branches over more layers strengthens the distinction between groups.

**Choice of channel dropping for MCD.** Our hypothesis for MCD lies in encouraging the concept channels of the model to contribute to the overall image representation with limited context. To validate this, we run experiments to identify the networks ($\mathcal{S}$ or $\mathcal{T}$) on which the channels need to be dropped in order to yield better representations. From Table 5, we observe that ensuring the concept channels are preserved during the forward pass results in better performance (None, $\mathcal{S}$ only, $\mathcal{S} + \mathcal{T}$), compared to the context-concept agnostic sampling strategies such as ChannelViT and DiChaViT. We also find that over no sampling at all, sampling the context under either setting ($\mathcal{S}$ only or $\mathcal{S} + \mathcal{T}$) yields better ID performance.

This observation aligns with our motivation of using MCD, where a masked context generally outperforms a non-masked context channel set. However, we observe a drop in average performance when the teacher context channels are masked ($\mathcal{S} + \mathcal{T}$). We attribute this to the limited learning capability caused by the weak teacher signal, as seen in existing SSL methods where masked teachers such as SdAE (Chen et al., 2022) under perform in linear evaluation tasks in comparison to methods with full teacher representations.

## 5 CONCLUSION

We introduce C3R, a two-fold architectural and pre-training framework that builds strong IHC image representations that transfer well to datasets with unobserved channel configurations. We build C3R based on the context-concept principle of IHC images, which we validate experimentally and show that this principle can be modeled and transferred across IHC datasets. We show that C3R significantly outperforms existing methods in both ID and OOD tasks, and matches dataset-targeted OOD training and evaluation strategies without any re-training. Overall, this work offers a new perspective on leveraging this principle in IHC datasets and opens a pathway for cross-dataset generalization without requiring dataset-specific adaptation or retraining at the image-level.

**Reproducibility statement** The summarized and detailed methodologies of C3R are outlined in Section 3.2 and Appendix A, respectively. The hyper-parameters used for pre-training and evaluation can be found in Appendix C. The code is available at https://anonymous.4open.science/r/C3R-5015.

**LLM usage statement** We use an LLM to aid and polish writing, mainly for grammar correction. We do not use an LLM for research ideation, code generation or finding related work. Therefore, we do not consider an LLM as a contributing author to this study.

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

APPENDIX

We organize the supplementary material of the main paper as follows. Appendix A describes in detail, the architecture of C3R. Appendix B provides further insight into design choices of C3R, and additional evaluations under channel-sparse settings. Appendix C describes datasets and implementation protocols used for training and evaluation, and code resources. Appendix D addresses the broader impact and limitations of this study.

## A  C3R - DETAILED METHODOLOGY

**Nomenclature.**   We use the following definitions for the upcoming sections.

$$
\begin{array}{ll}
x & : \text{multi-channel input image} \\
h_{c1}, h_{c2} & : \text{context and concept convolutional stem} \\
f_{c1}, f_{c2} & : \text{context and concept encoders} \\
g & : \text{combiner function} \\
f_s & : \text{joint encoder} \\
\mathcal{S}, \mathcal{T} & : \text{student and teacher network}
\end{array}
$$

### A.1  CCE ARCHITECTURE

**Grouped stem networks** $(h_{c1}, h_{c2})$**:**   For each group (context, concept), we initialize a convolutional stem network. Each stem is identical in architecture but contain group-dependent parameters. The stem network first consists of an instance-normalization for each channel in the group. The motivation for the instance-norm is to over-amplify high-intensity regions that occupy a small area in a given channel. We find that this leads to increased performance especially for OOD tasks.

After the instance-norm, each channel in a group undergoes a group-wise convolution that patchifies its channel images, using 1 input channel and $d$ output channels. These convolutions are equivalent to standard ViT patch embedding layers but operate on single-channel inputs. Both the context and concept groups thus produce tokenized feature maps, which are pre-pended with group-specific `cls` tokens, and then passed to their respective group-wise encoders.

Given the input $x = [x_{c1},\, x_{c2}]$, we apply the instance-norm and grouped convolutions, and prepend `cls` tokens as follows:

$$
\tilde{x}_{c1}^{i} = \big[\tilde{x}_{\texttt{cls},c1}, \texttt{reshape}\big(\texttt{conv}_{c1}\big(\texttt{IN}(x_{c1}^{i})\big)\big)\big] \in \mathbb{R}^{N \times d}, \quad i = 1, \ldots, C_1, \tag{15}
$$

$$
\tilde{x}_{c2}^{j} = \big[\tilde{x}_{\texttt{cls},c2}, \texttt{reshape}\big(\texttt{conv}_{c2}\big(\texttt{IN}(x_{c2}^{j})\big)\big)\big] \in \mathbb{R}^{N \times d}, \quad j = 1, \ldots, C_2, \tag{16}
$$

where $\texttt{IN}(\cdot)$ is the instance normalization, $\texttt{conv}_{c1}(\cdot)$ and $\texttt{conv}_{c2}(\cdot)$ are group-specific convolutional layers with equal kernel size and stride, with 1 input channel and $d$ output channels, and $\texttt{reshape}(\cdot)$ flattens the spatial dimensions into $N - 1$ tokens of dimension $d$. We prepend context and concept `cls` tokens to each tokenized channel.

After this step, each group is represented as a list of tokenized feature maps,

$$
\tilde{x}_{c1} = \big[\,\tilde{x}_{c1}^{1}, \tilde{x}_{c1}^{2}, \ldots, \tilde{x}_{c1}^{C_1}\,\big], \qquad \tilde{x}_{c2} = \big[\,\tilde{x}_{c2}^{1}, \tilde{x}_{c2}^{2}, \ldots, \tilde{x}_{c2}^{C_2}\,\big],
$$

where each element lies in $\mathbb{R}^{N \times d}$. Thus, the list length corresponds to the number of channels in the respective group, i.e. $|\tilde{x}_{c1}| = C_1$ and $|\tilde{x}_{c2}| = C_2$.

**Group-wise encoders** $(f_{c1}, f_{c2})$**:**   Each context and concept feature map is then passed to group-wise encoder layers. In this case, each encoder is simply a sequence of ViT blocks with dimension $d$. Each channel feature map in a group is passed through its group-specific layer independently, yielding intermediate representations per-channel per-group.

Given the input $\tilde{x} = [\tilde{x}_{c1}, \tilde{x}_{c2}]$, we pass each group through its encoder layers as follows:

$$\hat{x}_{c1}^{i,l+1} = \texttt{blk}_{c1}^{l}(\hat{x}_{c1}^{i,l}) \in \mathbb{R}^{N \times d},$$
$$i = 1, \ldots, C_1, \ l = 1, \ldots, L_c, \quad \hat{x}_{c1}^{i,1} = \tilde{x}_{c1}^{i}, \ \hat{x}_{c1}^{i} = \hat{x}_{c1}^{i,L_c+1} \tag{17}$$

$$\hat{x}_{c2}^{j,l+1} = \texttt{blk}_{c2}^{l}(\hat{x}_{c2}^{j,l}) \in \mathbb{R}^{N \times d},$$
$$j = 1, \ldots, C_2, \ l = 1, \ldots, L_c, \quad \hat{x}_{c2}^{j,1} = \tilde{x}_{c2}^{j}, \ \hat{x}_{c2}^{j} = \hat{x}_{c2}^{j,L_c+1} \tag{18}$$

Here, $\texttt{blk}_{c1}(\cdot)$ and $\texttt{blk}_{c2}(\cdot)$ denote the group-specific encoder blocks with hidden dimension $d$, and $L_c$ is the depth of each encoder. Each group is now represented as a list of intermediate representations,

$$\hat{x}_{c1} = \left[ \hat{x}_{c1}^{1}, \hat{x}_{c1}^{2}, \ldots, \hat{x}_{c1}^{C_1} \right], \qquad \hat{x}_{c1} = \left[ \hat{x}_{c2}^{1}, \hat{x}_{c2}^{2}, \ldots, \hat{x}_{c2}^{C_2} \right],$$

where each element lies in $\mathbb{R}^{N \times d}$. Thus, the list length corresponds to the number of channels in the respective group, i.e. $|\hat{x}_{c1}| = C_1$ and $|\hat{x}_{c2}| = C_2$.

**Combiner function ($g$):** We implement a simple aggregation mechanism to combine the intermediate representations. Specifically, we mean-pool each per-channel representation across a group, then concatenate the two group representations along the feature dimension. Finally, we pass the concatenated representation through a learnable layer-normalization.

Given the input $[\hat{x}_{c1}, \hat{x}_{c2}]$, we combine the representations using $g(\cdot)$ as follows:

$$\hat{x}_{c1,\texttt{pool}} = \frac{1}{C_1} \sum_{i=1}^{C_1} \hat{x}_{c1}^{i} \in \mathbb{R}^{N \times d}, \tag{19}$$

$$\hat{x}_{c2,\texttt{pool}} = \frac{1}{C_2} \sum_{j=1}^{C_2} \hat{x}_{c2}^{j} \in \mathbb{R}^{N \times d}, \tag{20}$$

$$\hat{x} = \texttt{LN}\big(\texttt{concat}\left(\hat{x}_{c1,\texttt{pool}}, \hat{x}_{c2,\texttt{pool}}\right)\big) \in \mathbb{R}^{N \times D}, \tag{21}$$

where $\texttt{concat}(\cdot)$ is the concatenation operation along the feature dimension, $\texttt{LN}(\cdot)$ is the layer normalization, and $D = 2d$ is the new feature dimension of the joint representation.

**Joint encoder ($f_s$):** The combined intermediate representation is passed through a sequence of ViT blocks of dimension $D$. Given the input $\hat{x}$, we pass the representations through $f_s$ to produce the final representation $y$ as:

$$\hat{x}^{l+1} = \texttt{blk}_{s}^{l}(\hat{x}^{l}) \in \mathbb{R}^{N \times D}, \quad l = 1, \ldots, L_c, \quad \hat{x}^{1} = \hat{x}, \tag{22}$$

$$y = \hat{x}^{L_s+1} \in \mathbb{R}^{N \times D} \tag{23}$$

$$y_{\texttt{cls}} = \hat{x}_{\texttt{cls}}^{L_s+1} \in \mathbb{R}^{D} \tag{24}$$

where $\texttt{blk}_s(\cdot)$ denote joint encoder blocks with hidden dimension $D$, and $L_s$ is the encoder depth. For downstream tasks, we extract $y_{\texttt{cls}}$, the cls token of the final representation.

## A.2 MASKED CONTEXT DISTILLATION (MCD)

Assuming we draw two global crops, the pseudo-code of Masked Context Distillation (MCD) is shown in Fig. 5. However, like DINO and iBOT, MCD can be extended to multiple crops, and in this study, we use 2 global crops and 10 global crops following iBOT.

## B ADDITIONAL RESULTS

### B.1 THE CONTEXT CONCEPT PRINCIPLE

**Validating the context-concept principle in IHC images - extended results** Table 6 shows the extension of Table 1, with the computed parity $P_c$ and entropy $H_c$ for each channel for each dataset.

```
# Inputs:                                    # Inputs:
# x : B x C x H x W tensor                    # u, v: B x (C1+C2) x H x W tensors (views 1 and 2)
                                              # S, T: student and teacher encoders
def sample_chans(x):                          # p_s, p_t: projection heads

    C = x.size(1) # num. channels             def forward_mcd(u, v, S, T, p_s, p_t):

    # Integer between 1 and C                      # Split crops into context and concept
    c = randint(1, C)                             u_c1, u_c2 = split_groups(u)
                                                  v_c1, v_c2 = split_groups(v)
    # sample c from the range of C
    idxs = sample(range(C), c)                    # Sample student context channels
                                                  u_c1_s = sample_chans(u_c1)
    # Sample channels                             v_c1_s = sample_chans(v_c1)
    x_ = x[:, idxs] # B x c x H x W
                                                  # Forward student
    return x_                                     yu_s = S(u_c1_s, u_c2, mask=True)
                                                  yv_s = S(v_c1_s, v_c2, mask=True)

         (a) Channel sampling.                    # Forward teacher
                                                  yu_t = T(u_c1, u_c2, mask=False)
                                                  yv_t = T(v_c1, v_c2, mask=False)

                                                  # Projection head
                                                  zu_s, zv_s = p_s(yu_s), p_s(yv_s)
                                                  zu_t, zv_t = p_t(yu_t), p_t(yv_t)

                                                  # Losses
                                                  L_mcd   = KLDivLoss(zu_s,zv_t)+KLDivLoss(zv_s,zu_t)
                                                  L_patch = iBOTPatch(zu_s,zu_t)+iBOTPatch(zv_s,zv_t)

                                                  loss = L_mcd + L_patch

                                                  return loss
```

(b) Forward pass with MCD loss given two crops $u, v$.

Figure 5: Pseudo-code for Masked Context Distillation. Here, `mask` denotes the masked image modelling component in iBOT. (a) shows the channel sampling, (b) shows the forward pass with student/teacher networks.

We observe a high parity for the context channels vs. the concept channels, and high entropy for the concept channels vs. the context channels. This validates our assumption that the context channels are statistically similar across instances in a dataset.

Table 6: Parity $P_c$ and entropy $H_c$ of cluster-assigned channel features in HPA and JUMP-CP.

| Group | Channel | HPA | | JUMP-CP | |
|---|---|---|---|---|---|
| | | $P_c$ | $H_c$ | $P_c$ | $H_c$ |
| Context | Microtubules | 0.942 | 0.374 | - | - |
| | Nucleus | 0.881 | 0.656 | 0.943 | 0.372 |
| | ER | 0.983 | 0.140 | 0.489 | 1.496 |
| Concept | Protein | 0.759 | 1.025 | - | - |
| | RNA | - | - | 0.217 | 1.987 |
| | AGP | - | - | 0.513 | 1.470 |
| | Mitochondria | - | - | 0.357 | 1.918 |

However, the parity of the ER channel in JUMP-CP is notably lower, and its entropy higher than expected for a context channel, although these metrics behave as expected in HPA. This is likely due to the overall entropy of JUMP-CP being higher than HPA, where JUMP-CP is relatively a noisy dataset compared to HPA which contains clear and high-resolution images. Such noise can blur the distinction between context and concept channels, leading to less clear clustering.

**Group assignments during evaluation**  Here, we outline the group assignments used during evaluation. Overall, we perform experiments with images from 3 datasets; HPA, JUMP-CP and WTC-11.

Table 7: Group assignments for HPA, JUMP-CP and WTC-11 datasets.

| Dataset | Context | Concept |
|---------|---------|---------|
| HPA | Nucleus, ER, Microtubules | Protein |
| JUMP-CP | Nucleus, ER | Mitochondria, Golgi Apparatus (AGP), RNA |
| WTC-11 | Nucleus | Protein, Membrane |

## B.2 ADDITIONAL EXPERIMENTS ON C3R

**Effect of layer depths for $f_c$ and $f_s$**  The design of CCE leads to exploring the amount of separate per-group processing needed for the context and concept channels before merging. This dictates the layer depths of $f_{c1}$ and $f_{c2}$. Moreover, as we adjust the layer depth of $f_s$ for CCE to maintain the total parameter count of a baseline ViT, the depth of $f_s$ indirectly reflects the amount of joint processing carried out after merging the context and concept groups.

To investigate the extent of per-group processing needed, we first vary the number of layers of $f_{c1}$ and $f_{c2}$, and keep the number of layers of $f_s$ adjusted to match the baseline parameter count.

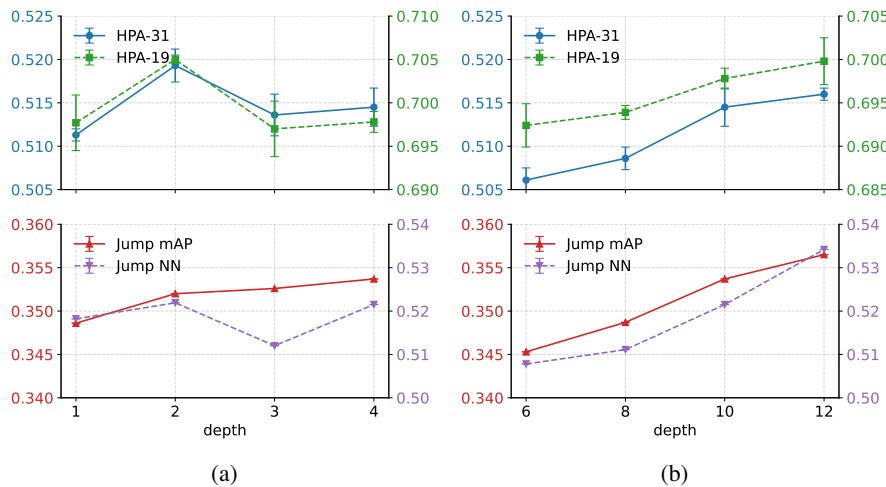

(a)                                                    (b)

Figure 6: Performance impact from (a) $f_c$ (b) $f_s$ depths on HPA-Loc (top) and JUMP-Ret (bottom). All experiments are performed on the CCE architecture with ViT-S and without Masked Context Distillation.

Fig. 6a shows that two layers each for $f_{c1}$ and $f_{c2}$ yields the best overall performance on HPA-Loc, whereas the effect of pre-merging depth on JUMP-Ret task is negligible. Given that the total parameter count is fixed, we investigate if the reduction in performance for HPA with higher depth for $f_{c1}$ and $f_{c2}$ is due to the loss of low-level feature granularity (caused by excessive abstraction before merging) or whether the adjusted number of combined layers of $f_s$ is insufficient to build a strong global representation.

To explore this, we fix the number of $f_{c1}$ and $f_{c2}$ layers at 4 (the deepest branch depth experimented previously) and vary the depth of $f_s$ from $[6, 8, 10, 12]$, with 12 being equivalent to the depth of a standard ViT backbone.

Fig. 6b shows the effect of the depth of $f_s$ given a fixed branch depth, where we observe that the performance increases with increasing depth for both HPA-Loc and JUMP-Ret. However, for HPA, the overall performance does not exceed that of a parameter-normalized network with two branched layers, implying that HPA benefits from preserving low-level features before merging. In contrast,

JUMP-Ret metrics show a global increase with increasing branch and shared depth, indicating that the JUMP-CP dataset benefits from both a deeper network due to more parameters, and a more abstract representation of groups with the use of deeper branches.

We suspect that this is mainly due to the difference in granularity between in datasets: HPA is inherently more fine-grained than JUMP-CP, hence HPA benefits from low level information even after merging, while JUMP-CP does not.

## B.3 ADDITIONAL EXPERIMENTS ON MCD

**Effect of context channel sampling rates for MCD** We explore the optimal sampling rate for the context channels when training with MCD. Specifically, we run fixed sampling rates where we drop $c = 1$ and $c = 2$ channels, and a variable sampling rate where we randomly drop $c \in \{0, 1, 2\}$ context channels, sampled uniformly at random. The total number of context channels in the HPA dataset is 3 (Microtubules, ER and Nucleus).

Table 8: Effect of context channel sampling rates in during MCD training. All experiments are performed on ViT-S.

| $c$ | HPA-Loc | | JUMP-Ret | |
|---|---|---|---|---|
| | 31-loc | 19-loc | mAP | kNN |
| Base-HPA | 0.505 | 0.686 | // | // |
| Base-CP* | // | // | 0.355 | 0.507 |
| 0 | 0.516 | 0.702 | 0.351 | **0.529** |
| 1 | 0.531 | 0.708 | **0.356** | 0.521 |
| 2 | 0.513 | 0.700 | 0.336 | 0.499 |
| $\{0, 1, 2\}$ | **0.536** | **0.722** | 0.354 | 0.518 |

Table 8 shows the effect of context channel sampling rates on ID and OOD performance. Here, we find that a random sampling rate $c = \{0, 1, 2\}$ yields better overall performance vs. fixed sampling. Consistently sparse sampling at $c = 2$ exhibits the lowest performance, possibly because the distillation task becomes too difficult. In contrast, dropping only a single channel at $c = 1$ yields similar OOD metrics but lower ID metrics. This result, along with the higher metrics obtained with variable sampling rates, suggests that the student $\mathcal{S}$ can still benefit from sparse context with $c = 2$, provided it receives enough support from other training iterations where fewer channels are dropped (e.g., $c = 0$ or $c = 1$).

**Ablation of MCD on CHAMMI-ZS and JUMP-TRex evaluations** Table 9 shows the effect of MCD on JUMP-CP evaluation using TRex pipeline, where we observe a marginal improvement with MCD on 5 out of 6 tasks. Moreover, Table 10 shows a significant improvement with MCD on the CHAMMI-ZS benchmark.

Table 9: Effect of MCD on JUMP-TRex.

| MCD | A549 | | U2OS | | A549+U2OS | |
|---|---|---|---|---|---|---|
| | Cmpd | MoA | Cmpd | MoA | Cmpd | MoA |
| ✗ | 0.441 | **0.348** | 0.338 | 0.291 | 0.323 | 0.335 |
| ✓ | **0.444** | 0.346 | **0.345** | **0.298** | **0.325** | **0.341** |

Table 10: Effect of MCD on CHAMMI-ZS.

| MCD | ID | OOD | CPS |
|---|---|---|---|
| ✗ | 0.634 | 0.379 | 0.338 |
| ✓ | **0.861** | **0.484** | **0.543** |

Overall, including the results from Table 4 in the main paper, we find that MCD improves performance on 3 out of 4 evaluations (HPA-ID, CHAMMI-ZS, JUMP-CP TRex) while matching the no MCD variant on a single evaluation (JUMP-CP retrieval). All better performing methods however contain some learnable component such as linear classifiers or learnable proxy tokens, while the JUMP-CP retrieval task does not. We aim to investigate the performance of pre-training strategies on pure zero-shot tasks without any dataset-specific adaptation in future work.

**Effect of MCD on limited context during evaluation**    Here, we study the translation of MCD learning from limited context during training, to inferring from limited context during evaluation. The HPA and JUMP-CP datasets contain three and two context channels, respectively. We perform evaluation on both datasets by dropping out a context channel, forcing the concept channels to contribute to the representation with limited context.

Table 11 shows the the effect of limited context channels during evaluation. We find that for HPA, when no MCD is applied, we there is a significant drop in performance. In contrast, MCD is robust to limited context, yielding very similar metrics with and without channel dropping.

However, MCD's robustness to channel dropping does not translate to OOD evaluation, as we observe a consistent drop in metrics with and without MCD, on JUMP-CP. Although we suspect that this is due to the noise in JUMP-CP causing the context representations with and without channel dropping to collapse into one representation, this warrants further investigation, which we aim to carry out in future work.

Table 11: Effect of limited context channels during evaluation. -[chan] corresponds to a given channel being dropped during evaluation. Arrows indicate average change relative to the "All" row for each method.

| MCD | Channels | HPA-Loc | | JUMP-TRex | |
|---|---|---|---|---|---|
| | | mAP-31 | mAP-19 | mAP | kNN |
| Base-HPA | | 0.505 | 0.686 | // | // |
| Base-CP* | | // | // | 0.355 | 0.507 |
| ✗ | All | 0.520 | 0.705 | 0.358 | 0.530 |
| | −[Nuc] | 0.515 | 0.697 | 0.352 | 0.529 |
| | −[ER] | 0.481 | 0.650 | 0.334 | 0.509 |
| | −[MT] | 0.494 | 0.679 | - | - |
| | −[Average] | 0.497 ↓**0.023** | 0.675 ↓**0.030** | **0.343** ↓**0.015** | **0.519** ↓**0.011** |
| ✓ | All | 0.536 | 0.722 | 0.354 | 0.518 |
| | −[Nuc] | 0.543 | 0.723 | 0.336 | 0.509 |
| | −[ER] | 0.527 | 0.713 | 0.339 | 0.501 |
| | −[MT] | 0.524 | 0.709 | - | - |
| | −[Average] | **0.531** ↓0.005 | **0.715** ↓0.007 | 0.338 ↓**0.016** | 0.505 ↓**0.013** |

## B.4    EXTENDED RESULTS

**Per-task results on CHAMMI-ZS**    Table 12 shows the detailed results on the CHAMMI-ZS benchmark. For each dataset, T1 is defined as the ID task while T2-4 are defined as OOD tasks. Across the CHAMMI-ZS benchmark, C3R consistently outperforms the single-channel baseline (Base-SC) and other channel-aware approaches. While Base-SC and existing methods show competitive performance on certain tasks (T2 and T3 on JUMP-CP), unlike C3R, they often struggle to maintain robustness across all datasets and task types.

Table 12: Per-task results on the CHAMMI-ZS benchmark. T{} denotes the task index within the dataset. For each dataset, T1 is defined as the ID task while T2-4 are defined as OOD tasks. Results are for ViT-S.

| Method | WTC-11 | | HPA | | | JUMP-CP | | | |
|---|---|---|---|---|---|---|---|---|---|
| | T1 | T2 | T1 | T2 | T3 | T1 | T2 | T3 | T4 |
| Base-SC | 0.758 | 0.560 | 0.904 | 0.806 | 0.368 | 0.797 | **0.570** | **0.189** | 0.069 |
| ChannelViT | 0.763 | 0.525 | 0.964 | 0.898 | 0.430 | 0.828 | 0.449 | 0.165 | 0.068 |
| DiChaViT | 0.709 | 0.513 | 0.957 | 0.869 | 0.425 | 0.770 | 0.405 | 0.167 | 0.079 |
| C3R | **0.770** | **0.602** | **0.978** | **0.944** | **0.605** | **0.835** | 0.530 | 0.142 | **0.083** |

## C DATASETS AND IMPLEMENTATION

### C.1 DATASETS

**Human Protein Atlas (HPA).**    We use the Human Protein Atlas (HPA) Subcellular Localization dataset (version 23) (Thul et al., 2017), a large-scale collection of immunofluorescence microscopy images capturing the expression and spatial distribution of 13,141 genes across 37 human cell lines. Each sample is stained with fluorescent markers targeting the nucleus, microtubules, endoplasmic reticulum, and a protein of interest, resulting in four-channel images that capture distinct structural and functional features of the cell. Protein localization is annotated across 35 subcellular compartments, covering both major organelles and fine-grained structures.

We extract single-cell instances using segmentation masks generated by the recommended HPA-Cell-Segmentation pipeline (Gupta et al., 2024). This yields a total of 1,138,378 single-cell crops. Data are split into training, validation, and test sets based on antibody identity to prevent information leakage across sets.

**JUMP-CP.**    In this study, we use a subset of the JUMP-Cell Painting (JUMP1) pilot dataset (Chandrasekaran et al., 2024). The full dataset includes approximately 300 million images of U2OS and A549 cells subjected to diverse chemical and genetic perturbations. The Cell Painting assay stains eight cellular components using six fluorescent dyes, imaged across five channels: nucleus, mitochondria, nucleoli and cytoplasmic RNA, endoplasmic reticulum, Golgi and plasma membrane, and actin cytoskeleton.

We select plates belonging to cell lines A549 and U2OS for analysis, where for JUMP-Ret evaluation we use U2OS following SubCell and for JUMP-TRex we use both cell lines following TRex. Each plate follows an identical treatment layout, consisting of 64 wells with negative controls and 320 wells treated with one of 302 unique compounds. Images from these plates are preprocessed and segmented into single-cell crops using DeepProfiler with default settings, following Subcell (Gupta et al., 2024).

Both HPA and JUMP-CP data are publicly available online[1][2].

### C.2 PRE-TRAINING

The C3R training pipeline can be found at: `https://anonymous.4open.science/r/C3R-5015`.

All single-cell crops are resized to $224 \times 224$ for HPA pre-training, $448 \times 448$ for HPA linear evaluation, and $112 \times 122$ for JUMP-CP. Inputs are $[0, 1]$ normalized per-channel, unless otherwise stated.

We pre-train all models using a Vision Transformer (ViT) backbone (Dosovitskiy et al., 2021), following the iBOT framework (Zhou et al., 2021)[3], on the HPA dataset (Thul et al., 2017). The dataset comprises $N_I = 1138378$ single-cell images annotated with $N_A = 11920$ antibodies. All cells with a unique antibody ID are distributed across 4 samples in the dataset. This results in an average of $N_I/(4 \times N_A) \approx 24$ cells per sample, and $4 \times N_A = 47680$ overall samples.

Following Gupta et al. (2024), we incorporate their antibody supervision loss, added directly to the iBOT objective without reweighting. Each iteration samples 8 cells per training sample, producing 8 positives and $B-8$ negatives for the supervised contrastive loss (where $B$ is the total batch size). The antibody loss and dataset utilities are obtained from the official SubCell repository[4].

For ViT-S/16 and ViT-B/16 encoders, we use AdamW optimizer (Loshchilov and Hutter, 2017) with a base learning rate of $5 \times 10^{-4}$, cosine annealing, and 10 warm-up epochs. ViT-S is trained for 100 epochs using a patch masking ratio sampled from $[0.1, 0.5]$ (mean $0.3 \pm 0.2$); ViT-B is trained for 50 epochs with masking ratio $[0.1, 0.7]$ (mean $0.4 \pm 0.3$). Each image undergoes 2 global crops and

---

[1]`https://www.proteinatlas.org`
[2]`https://github.com/broadinstitute/cellpainting-gallery`
[3]`https://github.com/bytedance/ibot`
[4]`https://github.com/CellProfiling`

10 local crops. We use batch sizes of 80 (ViT-S) and 48 (ViT-B) per GPU, training on 8 RTX 3090 GPUs.

For ChannelViT and DiChaViT (Bao et al., 2024; Pham and Plummer, 2024), we adapt their official model and loss implementations[5][6] to the iBOT pipeline. For both methods, we sample 2 channels per iteration. For ChannelViT, the same channels are sampled for both student and teacher networks. For DiChaViT, the first channel is randomly selected and shared between student and teacher networks, as the sampling of further channels occurs within the model framework.

For CCE and C3R (CCE + MCD), we define 2 branched layers per group at half the embedding dimension, leading to 11 shared layers with full embedding dimension (11 instead of the normal 12 to match the parameters of the baseline). In MCD, context channels are dropped from both global and local crop inputs to the student branch. All other settings match the base iBOT configuration.

## C.3 EVALUATION

**HPA-Loc**   We perform 19-class and 31-class multi-label classification on the field-of-view (FoV) level on the HPA dataset. A 3-layer MLP is trained for all evaluations. We use the official Subcell evaluation code (Gupta et al., 2024), which aggregates single-cell predictions to the field-of-view (FoV) level. Single-cell input size of 448 by 448 pixels was used during feature extraction prior to training the MLP. Models are trained using the Adam optimizer (Kingma, 2014) with a learning rate of $1e^{-3}$, Sigmoid Focal Loss (Lin et al., 2017), and early stopping based on validation mAP. Training is run for up to 100 epochs.

**JUMP-Ret**   For zero-shot evaluation on JUMP-CP, we extract embeddings from each single-cell image at a resolution of $112 \times 122$ pixels. The resulting embedding has dimensionality $3 \times d$ where $d$ is the model embedding dimension, as the Mitochondria, RNA, and AGP channels are each processed independently in combination with the Nucleus and ER channels, and their outputs are concatenated. The concatenated embeddings are then subjected to dimensionality reduction. To generate well-level profiles, we follow a two-stage aggregation strategy: first, cell-level embeddings are averaged to form Field-of-View (FoV) representations; second, FoV embeddings are averaged to produce well-level profiles. We report the k-NN evaluation and mAP metrics on replicate retrieval, where a positive replicate is subject to the same perturbation (out of 302 perturbations) as the query.

**JUMP-TRex**   For linear-evaluation on JUMP-CP, similarly to the zero-shot evaluation, we pre-compute cell-level embeddings of cell images from two cell lines; A549 and U2OS. The single-cell embeddings are passed through the TRex Adaptation module which contains a series of 4 fully connected layers with residual connections and GELU non-linearities, and a final output layer that predicts one of the 77 Mechanism-of-Action (MoA) classes. The MoA classification task is trained using focal loss and AdamW optimizer with a learning rate of $1e^{-3}$ and early stopping. After training, the output of the MoA classification layer and the third TRex adaptation layer are jointly used for both MoA prediction and compound matching, where compound matching is performed as JUMP-Ret.

**CHAMMI-ZS**   We use the official CHAMMI repository[7] and follow the benchmarking setup described in the original paper (Chen et al., 2023). Instead of the entire backbone and/or stem network, we pre-compute features for each IHC image in the CHAMMI dataset using the pre-trained network, and train a 2-layer MLP to optimize the proxy loss in CHAMMI. All methods are evaluated with the default CHAMMI hyper-parameter settings, with a learning rate of $5e^{-4}$.

**Overview of downstream tasks and their data distributions**   We summarize the downstream tasks in Table 13. Each task is defined by the feature adaptation strategy, the dataset used, and the distribution shift with respect to channels or tasks.

---

[5] https://github.com/insitro/ChannelViT
[6] https://github.com/chaudatascience/diverse_channel_vit
[7] https://github.com/chaudatascience/channel_adaptive_models

Table 13: Overview of downstream tasks and their data distributions. "OOD w.r.t channels" implies that the adaptation/evaluation is carried out on a dataset different to the source dataset (HPA), hence a different channel configuration in the evaluation dataset. "OOD w.r.t task" implies that within the evaluation procedure, the training (i.e.: linear classifier or proxy labelling, not direct retrieval), is performed on a different subset that is OOD by a different metric (such as cell line, or plate profile) other than channel configuration, within the main evaluation dataset.

| Evaluation | Frozen feature adaptation | Distribution | |
| --- | --- | --- | --- |
| | | **w.r.t channels** | **w.r.t task** |
| HPA-Loc | Fine-tuned with a 3-layer MLP for multi-label classification | ID: HPA | OOD: Novel antibodies |
| JUMP-Ret | Direct retrieval on frozen, normalized features | OOD: JUMP-CP | OOD: No training data |
| JUMP-Trex | Fine-tuned with a 4-layer MLP for compound and MoA prediction | OOD: JUMP-CP | OOD: Novel plates |
| CHAMMI-ZS | Frozen features with a 2-layer MLP. All ID tasks are trained jointly with learnable proxies. OOD tasks are solved via retrieval with the learned proxies. | OOD: WTC-11 | T1-ID: Known labels
T2-OOD: Novel labels |
| | | ID: HPA | T1-ID: Known cell lines
T2-OOD: Unseen cell lines
T3-OOD: Novel labels |
| | | OOD: JUMP-CP | T1-ID: Known plates
T2-OOD: Novel plates
T3-OOD: New CP dataset
T4-OOD: Novel labels |

## D  BROADER IMPACTS AND LIMITATIONS

**Broader Impact.** IHC imaging plays a critical role in clinical diagnostics and biomedical research, providing key insights into cellular morphology, protein localization, and disease progression at the subcellular level. However, the integration of deep learning methods towards these diagnostics has been limited by the lack of generalizable models that can operate reliably across heterogeneous datasets, where retraining models for every lab, institution, or imaging protocol is impractical and costly. Without dataset-specific adaptation or re-training, C3R has the potential to accelerate the integration of deep learning into diagnostic pipelines, reducing the time and cost of biomarker discovery, drug response prediction, and personalized treatment planning. However, as with all AI systems applied to biomedical data, misuse is possible. Malicious actors could potentially exploit representations from our pre-trained models for unauthorized diagnosis. This highlights the need of using models like C3R responsibly, with clear validation and compliance with ethical guidelines.

**Limitations.** Based on the context-concept principle, the ability to perform OOD evaluation on a target dataset depends on the principle being valid for the OOD dataset i.e: we find a natural separation of channels into context and concept in the target dataset. In the most common and publicly available IHC datasets (HPA (Thul et al., 2017), JUMP (Chandrasekaran et al., 2023), WTC-11 (Viana et al., 2023), OpenCell (Cho et al., 2022), Bridge2AI (Clark et al., 2024)) we find this assumption to be true, as the datasets have been created with the intention of detecting variations in channel intensities that then correspond to specific phenotypes. However, we have not explored IHC datasets that may not follow this principle. Furthermore, the CCE architecture serves as a proof of concept for our core hypothesis, that the context-concept principle in IHC datasets can be transferred to novel datasets without re-training. However, more elegant ways of building up the concept representation based on the context (e.g.: cross-attention (Bao and Karaletsos, 2023)) need to be explored. Also, like SubCell, DINO4Cells, ChannelViT and DiChaViT, we train and evaluate CCE on ViTs. We aim to adapt CCE towards building a general architectural framework beyond ViTs. Finally, we validate MCD using iBOT as our momentum-based SSL method. We aim to apply the idea of MCD to non-momentum based SSL methods in the future.

