# OpenReview forum: "C3R: Channel Conditioned Cell Representations for unified evaluation in microscopy imaging"
_ICLR.cc/2026/Conference — ICLR 2026 Conference Withdrawn Submission_

### Official Review · Reviewer_bZAw · 2025-10-30

**Soundness:** 2
**Presentation:** 2
**Contribution:** 2
**Rating:** 2
**Confidence:** 5

**Summary:**

The paper presents an architecture and a training framework to create microscopy image models that can be reused across different imaging configurations with different channels. The main idea of the architecture is to group channels in two sets: context channels and concept channels, which is biologically motivated by the way some imaging experiments are conducted. Context channels serve as a reference frame to observe variations in the information collected in concept channels.

The proposed method was trained on the HPA dataset and evaluated in out of distribution tasks. Out of distribution in this paper is focused on channels observed during training, which is a rather narrow definition. The presented experimental results indicate promising performance of the proposed approach, including evaluations in the JUMP dataset, and in the CHAMMI benchmark.

**Strengths:**

* The idea of grouping channels in context and concept is interesting, and biologically motivated.
* Based on this idea, the paper proposes an inductive bias to the architecture of ViTs and model training.
* The paper makes a good presentation of previous work.
* The experiments use established datasets to evaluate performance and investigate the properties of the models.

**Weaknesses:**

The paper has several issues and the experimental results do not support the conclusions and claims of this paper.

###  Context / Concept

* The main technical limitation of the proposed approach is that the definition of context and concept channels needs to be manually defined. The decision may be very arbitrary in practice.
* The quantification of context and concept seems to match the hypothesis, but it is not 100% supported by the data (as discussed in the Supplementary material). The separation can be artificial, and even if it’s quantitative, the paper is not presenting a solid method to automatically identify it in practice.

### Training
* The models and experimental evaluation are based on training with a single dataset (line 328).
* With a single training dataset, it is unclear how the model learns generalizable representations to other channel configurations. Specifically, the HPA dataset seems to have only one concept channel during training.
* The formulation of the model follows standard channel-wise token processing, originally proposed in the Channel-ViT paper and followed by others (Eq. 4 to 9). This architecture can be trained with varied numbers of channels. It is unclear why the proposed model is only trained with the fixed channels in HPA.
* It is not completely clear if the training algorithm is SSL or weakly supervised. There is no clear definition of the objectives and losses used in their models and the baselines.
* The focus on out-of-domain generalization is narrow to different channels only, when there are many other aspects of OOD generalization that can be studied. This includes different cell lines, treatments, batches and so on. The experimental designed is not clean in testing these types of OOD evaluations because of the restriction of training only on one dataset.

### Baselines

* Missing baseline evaluations. The Base-CP and Base-SC models are retrained with the same HPA data but with different channel settings to match the target dataset. Evaluation of models tailored to the channels and with the data of the specific task is important (specialized models).
* Missing baselines. Other pre-trained models such as OpenPhenom or DINOv2 were not included in the evaluation. The former is channel adaptive, and the other can be adapted as presented in Figure 2 (line 182). It is unclear why these models were not used.
* Baseline models seem suboptimal or impaired. ChannelViT and DiChaViT were trained with two channels at a time (line 1029). The models have the capacity to train with variable channel lengths, but the study is limiting this by arbitrarily choosing only 2, while the proposed models get to see 4 channels.
* DINO4Cells and SubCell were said to be the pre-trained available models (lines 350 and 351), but Table 2 indicates they are re-trained to match the JUMP-CP channel configuration. It is unclear how to interpret these results.
* Base-SC is trained with antibody loss, which is a supervised or weakly supervised loss (lines 328-332). It does not make sense to use a supervised loss with a single channel.
* The results in Table 3: the models are supposed to be trained with a 4 channel dataset, but the baselines only get to see 3 channels (line 331). In addition, the target dataset is 5 channels, so what channels are used and what channels are dropped is unclear, and it is an unfair comparison.

### CHAMMI benchmark

* CHAMMI is already a zero-shot benchmark, which uses K-NN evaluation for all 9 tasks (IID and OOD), and can be evaluated even with ImageNet pretrained models without any training. The statement that CHAMMI needs zero-shot style adaptation is incorrect, especially under the presented experimental setup (training only with HPA).
* The introduction of CHAMMI-ZS (zero shot) is highly misleading, because a specialized, two-layer MLP is actually being trained (lines 341-344). Zero-shot evaluation means no training, resulting in a conceptual and experimental mistake.
* The correct way of using CHAMMI for zero-shot evaluation is to compute features in all images and run the kNN evaluations. Those are the results that should be reported, specially if the paper claims generalization to unseen channels.


### Incorrect terminology
* Immunohistochemistry Channels => only one of the channels in HPA is IHC, the other channels are fluorescent channels. None of the channels in JUMP are IHC. This is the wrong terminology to refer to the datasets used in the presented experiments.
* Models do not have channels. The input images have channels, the models are functions that only have parameters. Statements such as “encourage the channels” (lines 093, 295) do not make sense, because the channels are fixed data points rather than parameters.
* In an experiment without re-training or fine-tuning, what gets frozen are the parameters of the model not the features (lines 334-340, 344, 424, etc).

### Other issues
* Figure 4 should keep together the bars of original and flip, rather than mAP and kNN. The quantities that need to be compared are split apart in the current layout. Also, the scale is misleading, giving the impression that the difference is large when it is actually very small.
* In Table 4, why is the Base-HPA worse than h_c alone in the HPA tasks? Is the baseline suboptimal? Base-HPA is supposed to get all the information needed to solve the HPA tasks in almost the same way as h_c. Is there any explanation for this difference?
* Figure 3 displays 5 channels (3 blue and 2 red) but in practice the experiments are all with 4 channels.
* Typo in line 203: $p_k^c$ when it should be $p_c^k$.
* Sloppy notation. In equation 8, the x parameters are missing the i and j super index respectively (e.g., $\hat{x}^i_{c1}$).

**Questions:**

* If the concept/ context idea is robust enough, why there is not a method to automatically detect the type?
* Why are all the experiments trained only on HPA? Does the method fail if trained on other datasets? This severely limits the potential of this study and the correctness of many reported experiments.
* Why is the method not trained with varied numbers of channels but rather with 4-fixed channels from HPA?
* Is there a limitation that prevents the model to be trained with disparate number of channels? In theory not, but why was this not tested?
* What is an antibody loss and what are the definitions of the training objectives? This is not clearly defined, and it is unclear whether the models are self-supervised or supervised.
* What does it mean to re-train baseline models DINO4Cells and SubCell to match the JUMP-CP channel configuration? When is this necessary?
* When dropping channels from the model (e.g., 3-channels for JUMP-CP), which ones are dropped and which ones are used and why?
* When adapting the proposed model trained with HPA, how are the assignments of context / concept channels made? Was this quantitative in CHAMMI, for instance? Where is this reported and why these choices?
* In the CHAMMI evaluation, is an MLP trained for each subset of the benchmark? The proposed model can be channel-adaptive, but an MLP is not. The details of how this is done are unclear.
* Why not conduct real zero-shot in CHAMMI? Why there is a need to train a two-layer MLP if the features are obtained with a frozen model? Why not just run the k-NN evaluation?

---

> ### Author Response · Authors · 2025-11-13
> **Clarification on training concerns**
>
> We thank reviewer bZAw for their valuable comments. We address the points raised in the review as follows.
>
> - **W1, Q1, Q8 : Context-concept**: Please refer to comment regarding the clarification on the context-concept principle.
> - **W2, Q2, Q3, Q4: Training**:
>   - **Pre-training on a single dataset (Q2, Q3, Q4)**: We pretrain on HPA because it is the only proteome-scale IF resource that pairs expert subcellular localization labels with a standardized four-channel confocal design (DNA/DAPI, microtubules, ER, plus protein-of-interest) at single-cell resolution, offering coverage of \>65% of protein-coding genes, diversity across 37 human cell lines, and consistent confocal acquisition. The HPA subcellular dataset is a collection of immunofluorescence images encoding the expression and spatiotemporal distribution of 13,141 genes in 37 cell lines; the lines span a broad range of tissues, cell types, ages, and sexes, providing the biological heterogeneity needed to build a strong pre-trained model. JUMP Cell Painting configurations emphasize perturbation phenotypes with lower-resolution epi-fluorescence and typically focus on two cell lines (e.g., U2OS and A549), which is excellent for benchmarking but quite narrow for training SSL models. WTC-11 represents a single hiPSC line geared to focused endpoints (e.g., cell cycle or intracellular organization), making it suitable for held-out evaluation domain rather than a proteome-wide training corpus. This combination of scale, gene coverage, cell-line diversity, high-quality confocal imaging, and rich metadata is precisely why the recent foundation models we compare against, SubCell and Dino4Cell, also uses HPA for pretraining.
>   - **Channel-wise token processing**:  ChannelViT and its variants use a fixed patch embedding layer for all channels, and then flattens all tokens along the token dimension, resulting in a long sequence length of $N\*C$ where $N$ is the token count per channel, and $C$ is the (sampled) channel count. Unlike ChannelViT and variants, we propose two separate lightweight encoder blocks for context and concept channel groups. For a context block, the block processes each context channel independently, resulting in a sequence length of $N$ per channel, and the context channel representations are then aggregated as in Eq.8 in the main paper. Although ChannelViT and their variants can handle multiple chanels, this is carried out by simply flattening individual channel features over the token dimension. This does not learn a context-concept aware representation from the input data, rather a joint ambiguous representation from all channels. On the other hand, CCE also handles a variable number of channels, but the channels are separated at the input stage depending on which group the channel is assigned to.
>   - **Training algorithm**: The baseline pre-training method is iBOT \+ antibody loss from SubCell. Therefore, for all methods, the pre-training is weakly supervised. This contains the masked image modelling (MIM) objective and the distillation objective (CLS) adopted from DINO, and the antibody loss from SubCell. In MCD, we replace the manner in which the distillation objective receives the student and teacher signals. As only the context channels are masked in the student, MCD can be categorized as weakly supervised objective, but the supervision is only the existing domain knowledge of which channels are context and concept in a dataset.
>   - **OOD being channels only**: We acknowledge that the main focus of OOD generalization in this paper is via unseen channels. However, the CHAMMI benchmark already contains the evaluations for unseen cell lines, drug perturbations and so on. In the CHAMMI benchmark, we outperform all methods on the CHAMMI OOD setting, where these task-specific OOD metrics are evaluated. We also outline how a task becomes OOD (i.e.: by channel or semantics) in Table 13 in the Appendix.

---

> > ### Author Response · Authors · 2025-11-13
> > **Clarification on baselines and CHAMMI**
> >
> > - **W3, Q6, Q7: Baselines**:
> >   - **Specialized model training**: We compare the OOD metrics on JUMP-TRex (Table 3\) and JUMP-Ret (Table 2\) using DINO4Cells, which trains a specialized model on HPA for HPA evaluation, and on JUMP-CP for JUMP-CP evaluations. Generally, models pre-trained on HPA tend to perform well on other datasets due to its richness. We observe this in JUMP-TRex and JUMP-Ret benchmarks, where SubCell (which is trained on 3-channel HPA) performs similarly to DINO4Cells.  We thank the reviewer for pointing this out, and will clarify that DINO4Cells for JUMP-CP is pre-trained on JUMP-CP in the next revision.
> >   - **OpenPhenom**: OpenPhenom is channel adaptive in the same breath as ChannelViT and their variants. Our work focuses on leveraging the context-concept principle to optimally adapt channels to target datasets, rather than being channel-adaptive by remaining channel-agnostic. Furthermore, OpenPhenom has been trained on a CellPainting dataset. Our results on SubCell and DINO4Cells shows that models trained on HPA perform similar on JUMP-CP, compared to models trained on JUMP-CP.
> >   - **DINOv2**: We use an ImageNet pre-trained DINOv2 encoder for Figure 2, as our aim is to highlight the textural features of channels and how they vary within the dataset.
> >   - **ChannelViT and DiChaViT being trained on 2 channels at a time**: These models have been designed for full-fine tuning on smaller datasets. Hence, the overall sequence length $N\*C$ matters less in that setting. However, in a pre-training setting, this incurs a large computational cost due to FLOP count. With 2 channels at a time, the FLOP count is approximately 4 times the training of any other baseline methods. Nevertheless, we train ChannelViT and DiChaViT for the same number of epochs as the baseline methods.
> >   - **Base-SC and antibody loss**: We thank the reviewer for pointing this out. We trained Base-SC with and without antibody loss, and found no difference in performance. Therefore, for consistency with other methods, we report the values for Base-SC trained with antibody loss.
> >   - **Table 3 results (Q6, Q7)**: SubCell trains the 3 channel model using Nucleus, ER, and Protein. For JUMP-Ret and JUMP-Trex, To obtain a JUMP-CP representation, SubCell passes three copies of the 5-channel JUMP-CP image; \[Nuc, ER, MITO\], \[Nuc, ER, AGP\], \[Nuc, ER, RNA\], where the last channel of each copy is matched to the Protein channel of HPA. Then, the three image representations are concatenated to form a $\\mathbb{R}^{3 \\times D}$ vector where $D$ is the embedding dimension. This vector is used for downstream analysis. Following SubCell, we implement Base-CP with iBOT and the antibody loss to achieve a 1-to-1 comparison with C3R.
> > - **W4, Q9: CHAMMI**:
> >   - **Zero shot terminology:** We acknowledge that the benchmark is not truly-zero shot and apologise for the lack of clarity. Our intention was to point out that the benchmark is zero-shot in terms of target channel configuration, as we are evaluating an HPA pre-trained model on JUMP-CP and WTC-11. In reality, it is a frozen-encoder evaluation, where the features (which are frozen and pre-trained using HPA), are evaluated on WTC-11, HPA, and JUMP-CP, where 2 of the 3 evaluation datasets are novel channel configurations. As the term ZS is misleading, we will rename the terminology to CHAMMI Frozen Encoder benchmark, as it is evaluated to measure the generalizability of pre-trained encoders in channel configuration X to a dataset with channel configuration Y.
> >   - **MLP (Q9)**: We train a joint MLP for all datasets, using the frozen encoder output CLS token as the MLP input. The MLP output feature is matched with the respective proxy token as in the original chammi benchmark.
> >   - **Real Zero-shot (Q10)**: In the original CHAMMI benchmark, proxy tokens (latent vectors) need to be trained using the ProxyNCA++ loss detailed in their paper. These proxy tokens act as label vectors, and are then matched with MLP outputs during inference.

---

> ### Author Response · Authors · 2025-11-13
> **Terminology and antibody loss**
>
> - **W5: Terminology**:
>   - **IF vs IHC**: We thank the reviewer for pointing this out. As all channels are a result of immunofluorescence microscopy, we will rename the terminology to IF images, datasets and channels.
>   - We have fixed the writing issues (lines 093, 295, 334, 344…) pointed out by the reviewer in the next revision.
> - **W6: Other issues**:
>   - **Base-HPA vs. h\_c only**: As $h\_{c1}$ and $h\_{c2}$ convolutional stems are group-specific, our hypothesis is that the convolutional stem projects the concept channels to a different vector space than the context channels. This is then carried on to the initial Q, K, V projections of the first joint ViT block, as the Q, K, V projection matrices itself now view the context dimensions as the first $D/2$ features, and the concept dims as the last $D/2$ features. This separation of context and concept is still carried on to the next few blocks until they are fully mixed.
>   - **Figure 3**: This is a conceptual figure to show the possible scenario with multiple concept channels. In the case of JUMP-CP forward pass, this is true. However, as correctly pointed out by the reviewer, for HPA it is only 1 concept channel. We will fix the figure accordingly.
>   - Typos and notation: We thank the reviewer for pointing this out. We have fixed these in the current revision.
> - **Q5: Antibody loss**:
>   - An antibody loss is the contrastive (InfoNCE) loss proposed in SubCell. Each cell image is stained with a unique antibody. During pre-training, images with similar antibodies are bought together in the feature space and images with different antibodies are pushed further apart. The detailed information for antibody loss can be found in the original paper. Although the antibody acts as a supervisory signal, it is not the protein localization label. Therefore, following SubCell, we class this as a weakly supervised algorithm.

---

### Official Review · Reviewer_WCdb · 2025-10-31

**Soundness:** 3
**Presentation:** 3
**Contribution:** 2
**Rating:** 4
**Confidence:** 3

**Summary:**

This paper introduces  C3R, a novel framework for representation learning fluorescence microscopy images. The authors identify an inherent context–concept structure in microscopy channels, where some channels (e.g., nucleus, ER) provide structural context while others convey experiment-specific information (typically about the protein under study). Building on this insight, C3R combines (i) a Context–Concept Encoder (CCE) that processes channel groups separately before joint integration, and (ii) a Masked Context Distillation (MCD) pretraining scheme that promotes robustness to missing or unseen channels. The method enables zero-shot generalization across datasets with novel channel configurations. Experiments on HPA, JUMP-CP, and CHAMMI-ZS benchmarks show that C3R outperforms prior methods (DINO4Cells, SubCell, ChannelViT, DiChaViT) both in in-distribution (ID) and out-of-distribution (OOD) settings.

**Strengths:**

1. Important and relevant problem of fluorescence microscopy representation learning
2. Overall well-written and easy to read.
3. Conceptually original idea (context/concept channel separation)
4. Comprehensive ablation study

**Weaknesses:**

1. Conceptually, it is not so clear how to distinguish between context and concept channels. First, this distinction depends on the study design: the same channel might provide context for one assay and content for another. For instance, nuclei are context information for subcellular localization screens, while they provide the relevant readout when studying cell cycle, nucleoli, aneuploidy or cell death.
2. Related to this, I am not entirely convinced by the analysis and results provided in section 3.1. First, the authors make no distinction between intra-image heterogeneity (if for instance localization patterns are less marked or feature higher inter-cell variability) and inter-image heterogeneity (if perturbations affect the localization more). Second, the authors show only summary metrics for $P_c$ and $H_c$, thereby occluding differences between context channels. Indeed, the nucleus channel might heavily influence both metrics, while the ER could be more similar to a content channel. Finally, whether a channel provides context or content depends on which biological process I am interested in (see my point 1). This is a problem, especially since, as the authors have demonstrated, switching channel categories further degrades performance.
3. The SSL MCD contribution is relatively incremental, and the validation (Table 5) sometimes depends on specific conditions, limiting robustness.
4. The work does not leverage recent state-of-the-art SSL methods such as DINOv2/3 or MocoV3, relying instead on iBoT, which may restrict comparative performance.
5. The approach lacks scaling to be considered truly generalizable; it appears more as a proof of concept than a fully deployable solution.
6. There appears to be some confusion between Immunohistochemistry (IHC) and Immunofluorescence (IF).
7. I would have expected a short explanation of the downstream tasks and biological objectives. Indeed, the datasets and associated tasks might not be known to the ICLR community.

**Questions:**

1. The authors use the term Immunohistochemical (IHC) throughout the text, which can be easily confounded with a modality used in digital pathology, usually not based on fluorescence. It would be better to use an unambiguous term, such as Immuno-fluorescence (IF) throughout the text.
2. Could the authors propose a systematic rule for assigning channels between context and concept?
3. Did the authors try SOTA SSL methods (e.g., DINOv2/3, MocoV3)? Is there a rationale for choosing iBoT?
4. In Figure 2, could the authors add a legend to facilitate the distinction between Concept and Context channels? Additionally, it is not very clear that Concept channels exhibit higher variance than Context channels from the UMAPs.
5. In Figure 4, it is noted that flipping the channels during inference degrades the metrics. Did the authors also train a model with flipped channels to assess the impact on performance?
6. Figure 6: Can the authors explain how the error bars are calculated, and why are they only displayed for the HPA-Loc dataset? Moreover, the  depth evaluation was performed by fixing the number of layers at 4. Can the authors justify this choice? Figure 6 seems to indicate that the optimal depth is 2.
7. Figure 6 seems to indicate that, overall, larger networks perform better. Have the authors tested more complex architectures to further improve performance?

---

> ### Author Response · Authors · 2025-11-13
>
> We thank reviewer WCdb for their valuable comments. We would like to address the reviewer's comments as follows.
>
> - **W1, W2, Q2: Distinguishing between context and concept channels:** Please refer to comment regarding the clarification on the context-concept principle.  https://openreview.net/forum?id=Faw8Jncudk&noteId=lsRyQKHt9X
> - **W3: Robustness of MCD**:
>   - We acknowledge that the retrieval performance on JUMP-CP (i.e.: JUMP-Ret task) is not improved by MCD in Table 4\. However, when frozen features are slightly updated via an MLP (i.e.: JUMP-Trex benchmark and CHAMMI benchmark), the performance increases with MCD (marginally for JUMP-Trex and significantly for CHAMMI), as seen in Table 9 and Table 10 in the supplementary material. Both JUMP-Trex and CHAMMI tasks contain some learnable component, but JUMP-Ret does not. We aim to investigate the performance of pre-training strategies on pure zero-shot tasks (i.e.: retrieval) without any dataset-specific adaptation in future work.
> - **W4, Q3: Use of iBOT over other SSL methods**:
>   - We use iBOT as it is a well established SSL methodology, that already outperforms MoCov3. Furthermore, DINOv2 is an adaptation of iBOT on a large-scale dataset, using the same class-wise and patch-wise losses, with the addition of incremental components outlined in Table 1 of their original paper \[1\]. CGSSL \[2\] has shown to improve over iBOT under low-data regimes (i.e.: ImageNet1k), but the method is very recent. Therefore, we stick to iBOT as our main pre-training pipeline.
> - **W5: Lack of scaling**:
>   - We understand that this method has been performed on ViT-S and ViT-B only. However, our model sizes are similar to other studies (ChannelViT, DiChaViT) in this domain in similar venues, and we also pre-train on a feature-rich dataset that has already been used as a pre-training dataset in SubCell and DINO4Cells. We hope to be able to scale this method to even larger models and datasets in the future.
> - **W6, Q1: Confusion between IHC and IF**:
>   - We acknowledge this issue and apologise for this oversight. For the final edit, as all channels are a result of immunofluorescence microscopy, we will change all terminology to immunofluorescence datasets (IF datasets), IF images, and IF channels.
> - **W7: Datasets and tasks**:
>   - We thank the reviewer for pointing this out. The detailed information on datasets and tasks is not present in the main paper, rather in the supplementary material. Detailed information of datasets can be found in Appendix C, and a summarized table of downstream tasks is shown in Table 13 in the supplementary material. We will incorporate this table into the main text in the final revision.
> - **Q4: Legend to facilitate the distinction between context and concept channels, and UMAP observation**:
>   - We will add the legend of context and concept channels to the figure as mentioned. From the UMAP, what we observe is that the channels in general show more similarities between instances, than between other channels of the same instance. This is unlike the default RGB case, where the data points of a UMAP for a given image will lie in the same location for both R, G, and B channels.
>   - Because it is not clear from the UMAP that the concept channels show higher variability, we perform the quantitative analysis in Table 1, and for the separate channels in Table 6 in the supplementary material. As the reviewer rightly mentions, the Nucleus channels shows the least variability overall. However, other context channels (in Table 6\) show lesser noise (lesser entropy) than concept channels in a given dataset, except for ER in JUMP-CP. Nonetheless, this analysis serves as a primer to understand the distinction between context and concept channels, but the main assignment of these channels should be biologically informed as mentioned in line 123 to 131 in the Related Work section of the paper.
> - **Q5: Flipping channels during inference**:
>   - The two group-specific lightweight encoders $f\_{c1}$ and $f\_{c2}$ are architecturally identical to each other. Therefore, flipping channels during training and keeping the original permutation of channel groups during inference is identical to training with the original permutation of channels and flipping them during inference.

---

> > ### Author Response · Authors · 2025-11-13
> >
> > - **Q6, 7: Error bars, depth evaluation and larger networks**:
> >   - The error bars are calculated for HPA-Loc for 5 random seeds (42-46) for the initialization of the MLP layers. As JUMP-Ret is a retrieval task, the frozen features remain the same and therefore no error bars are present.
> >   - In the first part of the depth evaluation, we set the total number of (separate \+ joint) layers to match the parameter count of a 12-layer vanilla ViT. Therefore, the initial search is on which combination of layer depths is optimal. The second part of the depth evaluation is to understand whether the truncation of the number of joint layers (by having more separate layers) is the main cause for the performance decrease. Or, whether the separate layers themselves being too deep is the main cause for performance decrease.
> >   - We find that, for HPA where the features are of higher quality compared to JUMP, the joint representation benefits from a lower-level feature map (i.e: 2 layers), while for JUMP-CP, the effect is negligible. But, post-layer depth has a larger effect on overall performance.
> >   - As correctly mentioned, the results in Figure 6 (b) show that increased depth generally yields better performance, indicating that model size may have an effect. For fairness, as mentioned before, we set layer depths such that the overall parameter count of the CCE encoder matches the parameter count of a 12-layer ViT encoder. Both SubCell and DINO4Cells have been trained with ViT-B backbones, while ChanneViT and DiChaViT mainly run their experiments on ViT-S backbones.
> >
> > \[1\] Oquab, Maxime, et al. "Dinov2: Learning robust visual features without supervision." arXiv preprint arXiv:2304.07193 (2023).
> > \[2\] S. Ahmed, J. Kittler, et al. "CG-SSL: Concept-Guided Self-Supervised Learning", NeurIPS 2025\.

---

### Official Review · Reviewer_FdJv · 2025-11-01

**Soundness:** 3
**Presentation:** 3
**Contribution:** 3
**Rating:** 4
**Confidence:** 4

**Summary:**

This paper addresses the challenge of applying deep learning to microscopy (IHC) images, which have inconsistent channel counts and configurations across datasets. This inconsistency normally prevents a model trained on one dataset from working on another without retraining.

The authors introduce the "context-concept principle," dividing channels into:
* Context: Stable structural references (e.g., Nucleus).

* Concept: Variable, experiment-specific information (e.g., Protein).

Based on this, they propose C3R, a framework with two parts:

* Context-Concept Encoder (CCE): A new architecture that processes these channel groups separately before merging them, enabling generalization to unseen channel configurations in a zero-shot manner.

* Masked Context Distillation (MCD): A training strategy that forces the model to learn from a limited set of context channels, improving representation robustness.

**Strengths:**

* The paper addresses a real issue in the field of image representation learning for microscopy images: modern image representation learning architectures are designed for natural RGB images, which always come in the three-channel format, where there is no significantly distinct information across channels.

* The paper introduces a new and intuitive way to structure microscopy data by grouping channels into "context" and "concept". The model's success should speak to the power of grouping channels accordingly.

* Using community-accepted benchmarks to evaluate results against other models

**Weaknesses:**

* The paper claims that its framework is the first of its kind to demonstrate strong zero-shot evaluation for this problem, but I found the following paper that purports to do the same, as is also similarly channel adaptive [1].

* The model's ability to evaluate a new OOD dataset relies on the assumption that the new dataset's channels can also be separated into "context" and "concept" groups. While the authors note this is true for the most common public IHC datasets (HPA, JUMP-CP, WTC-11, etc.), they have not explored datasets that may not follow this principle.


[1] L. Phillips and R. Donovan-Maiye, "CellRep: A multichannel image representation learning model," in Proceedings of the IEEE/CVF Conference on Computer Vision and Pattern Recognition (CVPR) Workshops, Jun. 2025, pp. 4312–4318.

**Questions:**

* Have you explored methods to learn this context-concept grouping automatically?

* MCD's success implies context channels are redundant, yet this robustness fails on the OOD JUMP-CP dataset. Does this mean the rules of context redundancy are not generalizable, and what does that imply for the 'context-concept principle' on new, unseen datasets?

---

> ### Author Response · Authors · 2025-11-13
>
> We thank reviewer Fdjv for their valuable comments. We would like to address the reviewer's comments as follows.
>
> - **W1: Comparison to CellRep**:
>   - CellRep proposes a channel encoder (lightweight ConvNet) to generate a channel-specific token, and then adds the channel-specific token to all patch tokens of that channel. This procedure is similar to existing works in ChannelViT and DiChaViT, which initialize latent channel tokens and add them to patch tokens. However, CellRep dynamically generates the channel token based on the input channel, but the core idea of channel-aware tokens remains the same to ChannelViT and DiChaViT. All patch tokens are then jointly passed through a ViT encoder (also similar to ChannelViT and DiChaViT), to then create a combined representation via average pooling over channels. This combined representation is then passed on to another larger ViT encoder as the student or teacher in a DINOv2 training setting.
>   - However, there is no material or sufficient information in the original study to reproduce CellRep. Nonetheless, C3R outperforms ChannelViT and DiChaViT in all tasks which share similar properties with CellRep.  The perfromance increase is mainly due to C3R viewing context and concept channels separately, as opposed to CellRep, ChannelViT or DiChaViT. We will include CellRep as a study on representation learning on IF datasets in the related work section of the final paper.
> - **W2: The context-concept assumption in IF datasets**: Please refer to comment regarding the clarification on the context-concept principle.  https://openreview.net/forum?id=Faw8Jncudk&noteId=lsRyQKHt9X
> - **Q1: Automatic grouping of context and concept**:
>   - We find from Table 1 in the main paper and Table 6 in the supplementary material, that there is a slight correlation between the inter-instance variability of a channel with its assigned group. However, as mentioned above, it is best to follow the biologists' protocol on what channels have been assigned to exhibit salient features to group channels when passing to the CCE encoder.
> - **Q2: MCD robustness fails on OOD JUMP-CP**:
>   - We acknowledge that the retrieval performance on JUMP-CP (i.e.: JUMP-Ret task) is not improved by MCD. However, when frozen features are slightly updated via an MLP (i.e.: JUMP-Trex benchmark and CHAMMI benchmark), the performance increases with MCD (marginally for JUMP-Trex and significantly for CHAMMI), as seen in Table 9 and Table 10 in the supplementary material. Both JUMP-Trex and CHAMMI tasks contain some learnable component, but JUMP-Ret does not. We aim to investigate the performance of pre-training strategies on pure zero-shot tasks (i.e.: retrieval) without any dataset-specific adaptation in future work.

---

### Official Review · Reviewer_6hTV · 2025-11-06

**Soundness:** 3
**Presentation:** 3
**Contribution:** 2
**Rating:** 4
**Confidence:** 4

**Summary:**

In this paper, the authors introduce a structured approach to training a channel adaptive model that can generalize well in zero-shot settings that are both in-distribution (having the same channels) and OOD (different channels during training and inference). They first introduce the Context-Concept principle by segregating channels into context channels (structural channels) vs concept channels (non-reference information). They propose Channel Conditioned Cell Representation (C3R) that incorporates the structural separation during training that generalizes well to ID and OOD at inference time. They also show their model performs state of the art on CHAMMI-zero-shot adapted dataset.

**Strengths:**

* Improved training strategy: Masked Context Distillation with concept context segregation is intuitive and adds biological data structure prior to training to an extent.
* Comprehensive evaluation and improved performance on several benchmark datasets (HPA, JUMP-CP, CHAMMI-ZS) in both in distribution and OOD settings.

**Weaknesses:**

* The assignment of concept or context groups are rather arbitrary or manually labeled subjectively.
* Random channel masking across Student/Teacher (i.e. channel masking as an augmentation baseline) is missing and the contribution of MCD (masking context channels specifically) is unclear.
* The contribution is very specific to IHC/fluorescence microscopy images and the subjective nature of context/concept selection makes it not generalizable and scalable to operate as a methodology.
 * CHAMMI-ZS benchmark doesn't seem to be truly zero-shot as there is a dataset specific MLP heads used for evaluating on the dataset.

**Questions:**

* What was the procedure for the segregation of context vs concept channels? It feels rather arbitrary than a specific methodology. The per group parity and entropy metrics do not necessarily capture the contextual or conceptual nature of the channels.
* Have you compared results on Masked Context Distillation with masking just concept channels or allowing masking all channels between student and teacher (Table 5 doesn't seem to answer that question clearly)? How much does Masking just the context contribute to performance compared to masking as an augmentation?

---

> ### Author Response · Authors · 2025-11-13
>
> We thank reviewer 6hTV for their valuable comments. We would like to respond to the reviewer’s comments as follows.
>
> - **W1, Q1: Assignment of context and concept groups**: Please refer to comment regarding the clarification on the context-concept principle.  https://openreview.net/forum?id=Faw8Jncudk&noteId=lsRyQKHt9X
> - **W2.1: Masking as an augmentation**:
>   - We have trained two models that perform group-agnostic channel masking/dropping, where channel masking can be viewed as an augmentation. The ChannelViT model uses hierarchical channel sampling (hcs), which is essentially sampling channels without replacement. During a forward pass, we sample a set of channels with the same channel indexes for both the student and teacher. Therefore, a concept channel can be dropped randomly. In addition to ChannelViT, we perform random sampling in the Base-SC. Here, we forward pass only one channel at a time, emulating an extreme case of channel masking. Here too, the concept channel can be dropped.  For both Base-SC and Channel-ViT, we find that sampling different sets of channels for both student and teacher affects training stability and downstream performance. Both Base-SC and ChannelViT show inferior results to C3R, indicating that preserving the concept channels is useful.
> - **W2.2, Q2: Masking context channels specifically:**
>   - CCE contains context and concept lightweight encoders (f\_c1) and (f\_c2)). Therefore, at any iteration, the model requires at least a single channel to pass from each of the lightweight encoders. In the HPA pre-training dataset, only a single concept channel exists. Therefore, only the context channels can be masked/dropped, and this makes it impossible to train the CCE model under random channel masking. However, we show, in Table 5, that masking context channels in either the student or teacher yields superior performance on 3/4 tasks as opposed to no masking. When channels are dropped randomly as an augmentation (Base-SC, ChannelViT) or without concept-awareness (DiChaViT), performance overall is much lower.
> - **W4: CHAMMI-ZS not being zero-shot:**
>   - We apologize for the lack of clarity on CHAMMI-ZS. Our terminology for this evaluation was defined by the zero-shot nature in terms of trained channels, as no training is performed specifically in learning the inter-channel relationship of the target channels in an encoder-level.
>   - In reality, it is a frozen-encoder evaluation, where the features (which are frozen and pre-trained using HPA), are evaluated on WTC-11, HPA, and JUMP-CP, where 2 of the 3 evaluation datasets are novel channel configurations. As the term ZS may be misleading, we will rename the terminology to CHAMMI Frozen Encoder benchmark, as it is evaluated to measure the generalizability of pre-trained encoders in channel configuration X to a dataset with channel configuration Y.

---

### Author Response · Authors · 2025-11-13
**Clarification on the Context-concept principle:**

The concept-context split is not arbitrary and not introduced by us. It reflects how immunofluorescence experiments are commonly designed and interpreted: (i) landmark/structural (context) channels that provide spatial reference for segmentation and registration; and (ii) readout (concept) channels whose signal is expected to vary with the biological factor of interest (antibody target, compound/MoA, genotype, pathogen). We adopt the roles declared by the assay; parity/entropy are reported only as consistency checks, not as a decision rule.


**1)  This practice is codified across canonical IF resources and datasets.**
-   Human Protein Atlas (HPA) [1]. Each sample pairs the protein of interest with three reference markers: nucleus (DNA/DAPI), microtubules (tubulin), and endoplasmic reticulum (ER; calreticulin), that serve as structural landmarks for interpreting localization.
-   Cell Painting / JUMP-CP [2]. The assay separates positional references (typically DNA and ER) from phenotype-bearing channels (RNA, AGP, mitochondria).
-   WTC-11 [3]. In cell-cycle assays, the nuclear/DNA channel is the readout (concept) because DNA content/organization encodes G1/S/G2/M; auxiliary stains (membrane/cytoplasmic markers or transmitted-light) are context for delineating cells.
-   CM4AI-Bridge2AI (ICC-IF) [4]. Proteins of interest are labelled with specific antibodies (readout), while cells are co-stained with DAPI (nuclei), anti-tubulin (microtubules), and anti-calreticulin (ER) as structural/positional markers (context)
-   Rabies diagnostics: The direct fluorescent antibody test detects rabies nucleocapsid antigen on fixed tissue/monolayers (the readout) and is interpreted relative to the structural background/counterstain (the context)
-   Across HPA, Cell Painting/JUMP-CP, WTC-11, CM4AI, and diagnostic IF, biologists predefine channel roles to ensure interpretability. Our paper exploits this established convention; we are not redefining which channel is which.

**2) Task dependence is expected and supports our framing**
-   Protein localization (e.g., HPA): nucleus/DAPI, microtubules, and ER provide context to situate the protein channel.
-   Morphological profiling (Cell Painting/JUMP-CP): DNA/ER act as context, whereas RNA/AGP/mitochondria are readouts carrying perturbation phenotypes.
-   Cell-cycle assays (e.g., WTC-11 cell-cycle): DNA is the readout, and structural labels (membrane/cytoplasm/brightfield) provide context.
-   We follow the assay’s intent rather than imposing a universal label. The fact that roles can flip across assays is a property of biology and study design, not evidence of arbitrariness.


**3) Manual is not arbitrary; it means assay-specified by domain experts**

-   Manual here means roles are specified a priori in protocols and dataset documentation (e.g., HPA’s four-channel schema; Cell Painting’s fixed five-channel design; rabies FAT antigen vs. background) to ensure reproducibility and interpretability. These conventions have been validated over years of use.


**4) Parity/entropy are sanity checks, not the basis of assignment**

-   Landmark channels typically appear more stable/unimodal, while readouts vary by design. We include parity/entropy to corroborate the expected roles; they do not define them. The definition comes from the assay’s biological specification.

**5) What happens when a new IF dataset arrives?**

-   We apply a biology-first policy that mirrors standard practice:
-   If the protocol/SOP designates a stain as a reference/landmark, we treat it as context.
-   If it is a target/readout stain, or documentation is ambiguous, we treat it as concept and validate via ablations to show conclusions are stable.

**6) Why our approach avoids per-task re-training (contrast with prior models)**
-   Many recent microscopy foundation/SSL models learn strong representations but do not explicitly exploit the landmark-vs-readout principle; as a result, they are typically trained per task. DINO4Cells and SubCell have to train separate models for task, whereas C3R uses one model and routes information by biological role (context vs. concept) that is already defined by the assay. This is the core practical advantage: no separate model per task is needed precisely because we exploit the universal, assay-level role assignment that biologists specify.

References:

[1] Thul, Peter J., et al. "A subcellular map of the human proteome." _Science_ 356.6340 (2017): eaal3321.

[2] Chandrasekaran, Srinivas Niranj, et al. "JUMP Cell Painting dataset: morphological impact of 136,000 chemical and genetic perturbations." _BioRxiv_ (2023): 2023-03.

[3] Viana, Matheus P., et al. "Integrated intracellular organization and its variations in human iPS cells." _Nature_ 613.7943 (2023): 345-354.

[4] Rincon, John, et al. "Bridge2AI: Building A Cross-disciplinary Curriculum Towards AI-Enhanced Biomedical and Clinical Care." _arXiv preprint arXiv:2505.14757_ (2025).

---

### Note · Authors · 2025-11-14

I have read and agree with the venue's withdrawal policy on behalf of myself and my co-authors.